EMBO
*reports*

# Hypoxia and loss of PHD2 inactivate stromal fibroblasts to decrease tumour stiffness and metastasis

Chris D Madsen[1,2,*], Jesper T Pedersen[2], Freja A Venning[2], Lukram Babloo Singh[2], Emad Moeendarbary[3], Guillaume Charras[4,5], Thomas R Cox[2], Erik Sahai[1,**] & Janine T Erler[2,***]

## Abstract

Cancer-associated fibroblasts (CAFs) interact with tumour cells and promote growth and metastasis. Here, we show that CAF activation is reversible: chronic hypoxia deactivates CAFs, resulting in the loss of contractile force, reduced remodelling of the surrounding extracellular matrix and, ultimately, impaired CAF-mediated cancer cell invasion. Hypoxia inhibits prolyl hydroxylase domain protein 2 (PHD2), leading to hypoxia-inducible factor (HIF)-1α stabilisation, reduced expression of αSMA and periostin, and reduced myosin II activity. Loss of PHD2 in CAFs phenocopies the effects of hypoxia, which can be prevented by simultaneous depletion of HIF-1α. Treatment with the PHD inhibitor DMOG in an orthotopic breast cancer model significantly decreases spontaneous metastases to the lungs and liver, associated with decreased tumour stiffness and fibroblast activation. PHD2 depletion in CAFs co-injected with tumour cells similarly prevents CAF-induced metastasis to lungs and liver. Our data argue that reversion of CAFs towards a less active state is possible and could have important clinical implications.

**Keywords** cancer-associated fibroblasts; hypoxia; PHD2; tumour invasion and metastasis; tumour stiffness
**Subject Categories** Cancer; Cell Adhesion, Polarity & Cytoskeleton; Signal Transduction

## Introduction

The tumour microenvironment plays an important role in the development and spread of cancer [1]. One of the most abundant cell types of the tumour microenvironment is cancer-associated fibroblasts (CAFs) [2]. CAFs comprise various fibroblast populations of distinct cellular origins that become activated during tumorigenesis, that is by cancer cells. CAFs are associated with tumour cells at all stages of cancer, and they support cancer progression by secreting chemokines, growth factors and various extracellular matrix (ECM) components, including collagen I [3–5]. CAFs are also responsible for the physical remodelling of the ECM and the subsequent dissemination of tumour cells from the primary tumour [6]. Indeed, the ability of fibroblasts to remodel collagen I has been shown to mediate malignant transformation [7] and metastatic outgrowth in breast cancer [5,8].

Hypoxia is another key feature of the tumour microenvironment [9]. Hypoxia is associated with metastasis and poor prognosis [10]; however, while all solid tumours over several millimetres contain regions of hypoxia, most tumour cells are actually non-migratory. Thus, the relationship between hypoxia and metastasis is not fully understood. The oxygen-sensing prolyl hydroxylase domain proteins (PHD1–3/EGLN1–3) are enzymes that hydroxylate and target the alpha-subunit of hypoxia-inducible transcription factors (HIFs) for degradation [11]. When oxygen tension drops, PHDs become less active, leading to the stabilisation of HIF proteins [12]. This subsequently drives an adaptive response inducing angiogenesis and metabolic reprogramming [13]. HIFs are generally considered as tumour-promoting transcription factors, and accordingly, targeted deletion of HIF-1α in malignant epithelial cells, endothelial cells or myeloid cells has been shown to suppress tumour growth [14–16]. Surprisingly, targeted deletion of HIF-1α in stromal fibroblasts enhances mammary tumour progression by reducing vascular density and myeloid cell infiltration [17]. In addition, conditional loss of PHD2 in endothelial cells and the simultaneous loss in T lymphocytes and myeloid cells have been shown to impair tumour progression through the regulation of HIF [18,19]. The perplexing role of HIF regulation in

1 Tumour Cell Biology Laboratory, The Francis Crick Institute (formerly Cancer Research UK London Research Institute), London, UK
2 Biotech Research and Innovation Centre (BRIC), University of Copenhagen, Copenhagen, Denmark
3 Department of Biological Engineering, Massachusetts Institute of Technology, Cambridge, MA, USA
4 Department of Cell and Developmental Biology, University College London, London, UK
5 London Centre for Nanotechnology, University College London, London, UK
*Corresponding author. Tel: +45 35 32 56 47; Fax: +45 35 32 56 69; E-mail: chris.madsen@bric.ku.dk
**Corresponding author. Tel: +44 20 7269 3165; E-mail: erik.sahai@crick.ac.uk
***Corresponding author. Tel: +45 35 32 56 66; Fax: +45 35 32 56 69; E-mail: janine.erler@bric.ku.dk

stromal cells encourages us to investigate the role of hypoxia on CAF-induced ECM remodelling and cancer invasion. Here, we show that hypoxia and loss of PHD2 revert CAF activation. This reversion is associated with loss of matrix stiffening and consequently reduction in spontaneous metastases to lungs and liver. We also demonstrate that PHD inhibitors decrease primary tumour stiffness as well as metastatic tumour burden. These findings support the use of PHD inhibitors in the clinic and emphasise how the tumour microenvironment can have seemingly opposing effects on cancer versus stromal cells.

## Results

### Hypoxia suppresses CAF-induced matrix remodelling

The direct influence of hypoxia on CAF activation and function is still poorly understood. To investigate this, we incubated human head and neck CAFs (HN-CAFs) and vulval CAFs (V-CAFs) in 3D collagen I matrices and evaluated their cell morphology. In order to investigate the specific role of CAFs under hypoxia, and not the function of low oxygen levels on collagen polymerisation, the 3D collagen gels were prepared under normoxic conditions. Once the gels had set, they were transferred to hypoxia (1% $O_2$) and compared to normoxia after 72 h of incubation. As shown in Fig 1A, hypoxic conditions induced an elongated spindle-like morphology of CAFs in 3D. We then turned our attention to the ability of CAFs to contract and remodel their surrounding matrix. Hypoxia and the hypoxia mimetic agent dimethyloxallyl glycine (DMOG) suppressed HN-CAF- and V-CAF-induced contraction of 3D collagen I matrices to levels comparable to the inhibition of ROCK (Y-27632) and non-muscle myosin II (blebbistatin), known to be key players in actomyosin contractility (Fig 1B–D). Shear rheology revealed that the reduced contraction of collagen gels under hypoxic conditions was also associated with reduced matrix stiffness (Fig 1E and F). Collagen stiffening can be caused by increased covalent cross-linking by enzymes of the lysyl oxidase (LOX) family [7,20–22]. However, we did not observe changes in collagen oligomerisation status under hypoxic conditions—as assessed by immunoblotting analysis of non-denatured proteins separated in Tris–acetate gels (Fig EV1A) [23]. Further, treatment with LOX family inhibitor *beta*-aminopropionitrile (β-APN) and five LOXL2 inhibitors (PXS1_1-5) did not alter CAF-induced collagen contraction (Fig EV1B and C). These

observations led us to look for alternative explanations for the altered collagen stiffness that we observe at low oxygen tension. CAF-induced matrix stiffening is also dependent on contractile actomyosin function (Fig 1C and D). Significantly, CAFs in hypoxia exhibited lower levels of markers of actomyosin contraction, including phospho-S19-myosin light chain (pMLC/pMYL9) and phospho-T696-MYPT (pMYPT/pPPP1R12A) (Fig 1G). The reduced levels of pMLC and pMYPT were also demonstrated using hypoxia mimetic agents (Fig 1H). All the above-mentioned effects were not due to loss of proliferation (Fig EV1D).

### Hypoxia suppresses CAF-induced cancer cell invasion

We have previously shown that force-mediated contraction and matrix remodelling by CAFs is required for the collective invasion of the squamous cell carcinoma (SCC) [6,24]. Upon CAF-mediated SCC invasion, the fibroblasts generate tracks through the matrix, which enable the SCCs to collectively invade [6]. In order to test the hypoxic effects on the CAF population alone, and not the SCC's, we modified the previously described organotypic invasion assay [6]. In this experimental set-up, the CAFs were allowed to remodel the 3D collagen matrix for 4 days under either normoxic or hypoxic conditions. CAFs were then killed and removed, and SCCs are placed on top of the gels and allowed to invade into the matrix for another 5 days under normoxic conditions (Fig EV1D). As shown in Fig 2A and B, hypoxia clearly decreases the ability of CAFs to drive SCC invasion. These results were further confirmed using other cancer cell lines: A431 and HT-1080 (Fig EV1E and F). Likewise, treatment with DMOG during the initial matrix remodelling period also inhibited CAF-induced SCC invasion (Fig 2C).

These data may seem surprising, as hypoxia is typically thought of as promoting cancer cell invasion; however, under the experimental conditions, the surrounding matrix is very dense and rigid and does not easily allow cancer cells to invade without the assistance of CAFs. As a matter of fact, highly invasive HT-1080 cells are still able to invade these matrices without the assistance of CAFs, and under such conditions, low oxygen tension actually promoted their invasion (Fig EV1F, right panels and zooms). These data support the notion that hypoxia induces cancer cell invasion, if the surrounding matrices are penetrable, but also reveal that CAFs lose their ability to remodel their environment under low oxygen tension due to decreased isometric tension.

▶

**Figure 1. Hypoxia suppresses CAF-induced matrix remodelling.**

A    F-actin staining of human HN-CAFs and V-CAFs grown in 3D collagen I/Matrigels. The images are maximum projected for visualisation. Bars show quantification of the longest distance of HN-CAFs grown in 3D collagen I/Matrigels, and of V-CAFs plated on top of gels. Bars represent mean ± s.d. *n* = individual cells from three experimental repeats. \*\**P* < 0.01; \*\*\**P* < 0.001; unpaired Student's *t*-test. Scale bar: 50 μm.

B    CAF-induced contraction assay. Images show HN-CAF-induced contraction of collagen I/Matrigels after 72 h of remodelling under normoxia and hypoxia.

C, D    Scatter plots show quantification of HN-CAF (C)- and V-CAF (D)-induced contraction relative to normoxia. Each data point represents an independent experiment. Line and error bars indicate mean ± s.d. \*\*\**P* < 0.001; one-way ANOVA test.

E, F    Matrix stiffness under normoxia and hypoxia. The storage modulus (G') of collagen I/Matrigels was measured by shear rheology after 72 h of CAF remodelling. CAFs were still alive within gels upon measurement. (E) Storage modulus was always measured over a decade of oscillation strain from 0.2 to 2%. (F) Scatter plot shows storage modules of matched experiments. Each couple of match data points represents an independent experiment. \*\**P* < 0.01; paired Student's *t*-test (two-tailed).

G, H    Immunoblotting analyses of the contractile machinery in HN-CAFs after 72 h under normoxia and hypoxia (G) and 72 h after treatment with hypoxia mimetic agents (H). Bars show quantification of mean ± s.d. phospho-myosin light chain levels normalised to actin levels. *n* = 4 experimental repeats for hypoxia vs. normoxia. *n* = 2 experimental repeats for hypoxia mimetic agents.

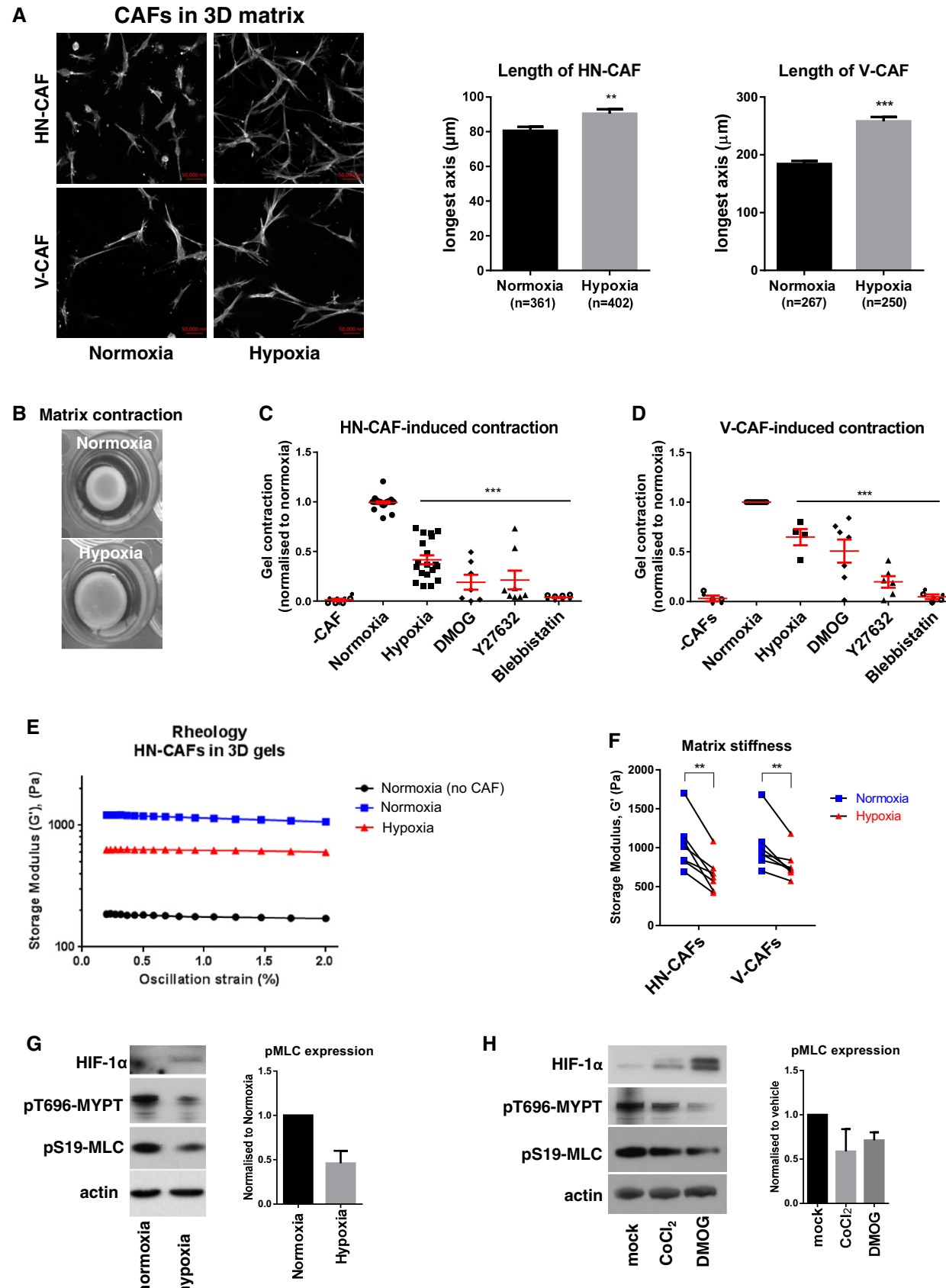

Figure 1.

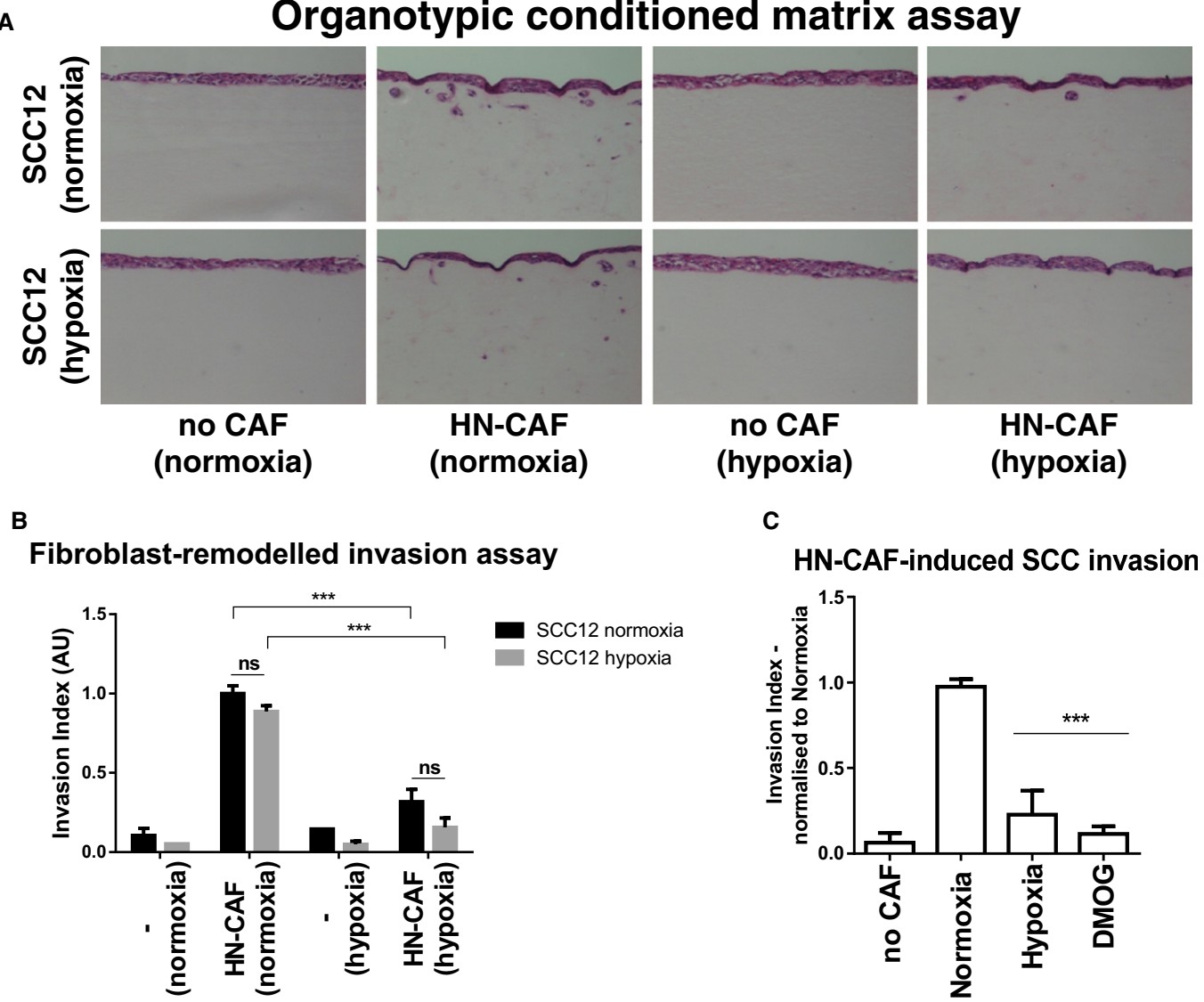

**Figure 2. Hypoxia suppresses CAF-induced cancer cell invasion.**

A  Organotypic invasion assay. H&E-stained sections of SCC12 cells cultured in the organotypic system in the absence or presence of HN-CAFs, and under normoxic and hypoxic conditions.

B  SCC12 invasion. Bars show quantification of SCC12 invasion in normoxia and hypoxia. Bars represent mean ± s.d. *n* > 3 experimental repeats. ns, non-significant; ***P < 0.001; unpaired Student's *t*-test (two-tailed).

C  Bars show quantification of SCC12 invasion relative to normoxia. Bars represent mean ± s.d. *n* > 3 experimental repeats. ***P < 0.001; unpaired Student's *t*-test (two-tailed).

## Hypoxia suppresses CAF activation

The origin of and molecular events driving the genesis of CAFs are areas of intense research, and it is assumed that CAFs maintain their activated phenotype presumably through acquisitions of genetic and/or epigenetic alterations in these cells [25–28]. On the contrary, very little information exists to support the deactivation and de-differentiation of CAFs towards a less aggressive phenotype. Our data strongly suggest that sustained hypoxia deactivates CAFs. We therefore tested whether hypoxia affected the expression levels of various CAF markers and ECM molecules (Fig 3A). Vimentin and PDGFRα

and PDGFRβ expression were unaffected by hypoxia; however, periostin (POSTN) and alpha-smooth muscle actin (αSMA/ACTA2) were consistently reduced after 48 h of hypoxia and after DMOG treatment in both HN-CAFs and V-CAFs (Figs 3A and B, and EV2A). This downregulation increased progressively over time (Fig 3C and D). Western blot analysis revealed that the reduced αSMA expression slightly preceded diminished pMLC levels (Fig 3B and D). Interestingly, flow cytometry analysis revealed that the reduction in αSMA was not caused by the appearance of an αSMA-negative subpopulation of CAFs, but was the result of reduced expression in all cells (Fig 3E). Similar data were obtained for pMLC (Fig 3E).

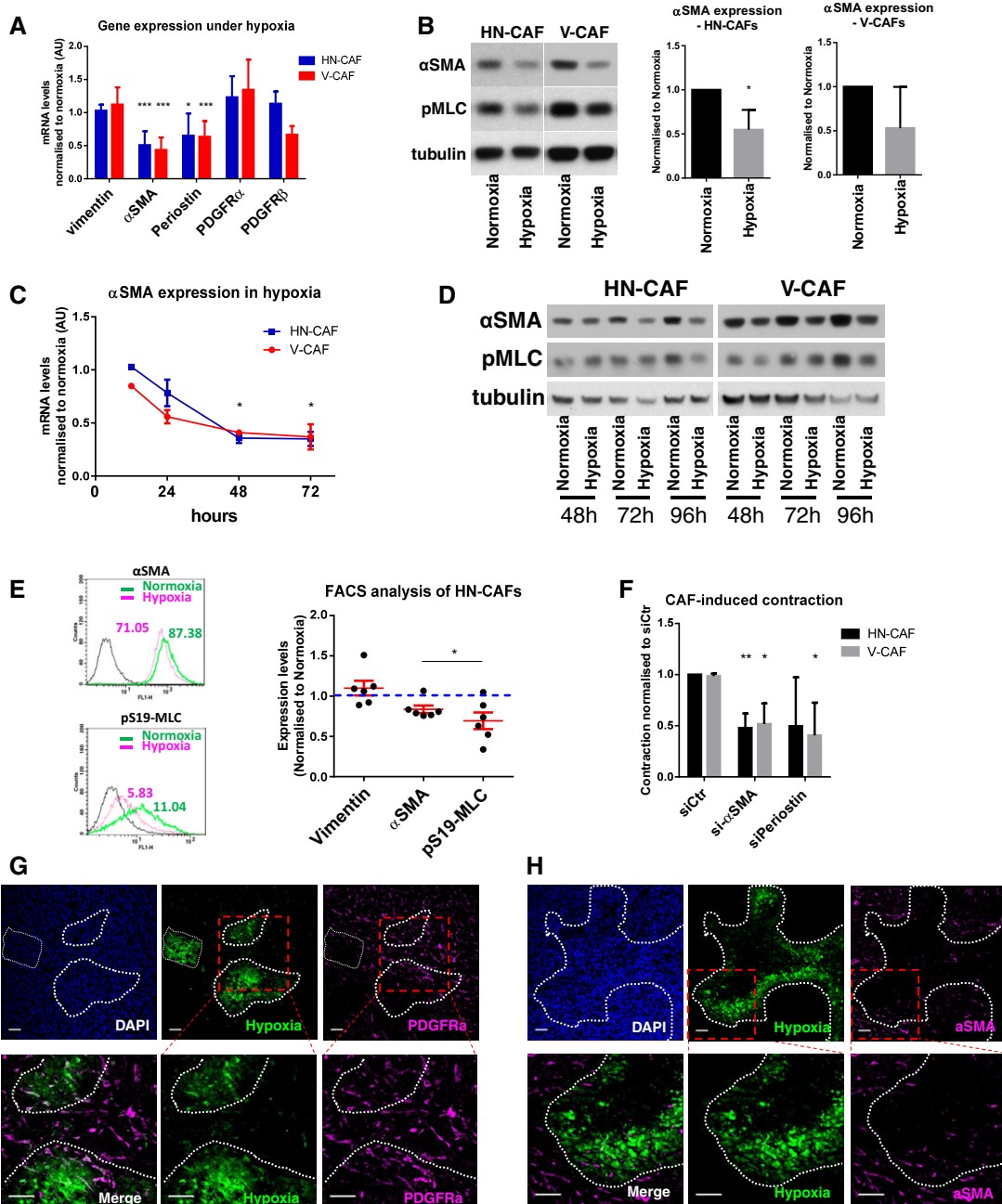

**Figure 3. Hypoxia suppresses CAF activation.**

A    CAF markers in hypoxia. Bars show mean ± s.d. mRNA levels quantified by qPCR after 48 h of hypoxic incubation. mRNA levels are normalised to normoxic levels. CAFs were plated on collagen I gels. n > 3 experimental repeats. *P < 0.05; ***P < 0.001; unpaired Student's t-test (two-tailed).

B    Immunoblotting analyses of αSMA and phospho-S19-MLC levels after 72 h in normoxia and hypoxia. CAFs were plated on gels. Bars show quantification of mean ± s.d. αSMA protein levels normalised to tubulin levels. n > 3 experimental repeats. *P < 0.05; unpaired Student's t-test (two-tailed).

C    Time course of αSMA expression in hypoxia. Graph shows the mean ± s.d. mRNA levels of αSMA quantified by qPCR. mRNA levels are normalised to normoxic levels. *P < 0.05; unpaired Student's t-test (two-tailed). n = 2 independent experiments each done in triplicate.

D    Time course of αSMA and phospho-S19-MLC protein levels in CAFs incubated under hypoxia.

E    Flow cytometry of αSMA and phospho-S19-MLC. HN-CAFs were subjected to flow cytometry analyses after 72 h in normoxia and hypoxia. Mean fluorescence intensity is shown from one representative experiment. Scatter plot shows quantification of αSMA and pS19-MLC levels relative to normoxia. Each data point represents an independent experiment. Line and error bars indicate mean ± s.d. *P < 0.05; unpaired Student's t-test (two-tailed).

F    siRNA-transfected CAFs were subjected to contraction assay. Bars shows mean ± s.d. quantification of CAF-induced contraction relative to normoxia siRNA control cells (siCtr). n > 3 experimental repeats. *P < 0.05; **P < 0.01; unpaired Student's t-test (two-tailed).

G, H    BALB/c mice were orthotopically injected with 4T1 breast cancer cells. After 3 weeks of tumour growth, mice were i.p. injected with the hypoxyprobe pimonidazole and sacrificed 1 h later. The primary tumour was then excised and stained for hypoxia, PDGFRα (G) and αSMA (H). The dotted lines represent the border of the hypoxic region. Lower panels represent the zoomed area depicted in red. Scale bar: 50 μm.

**A**

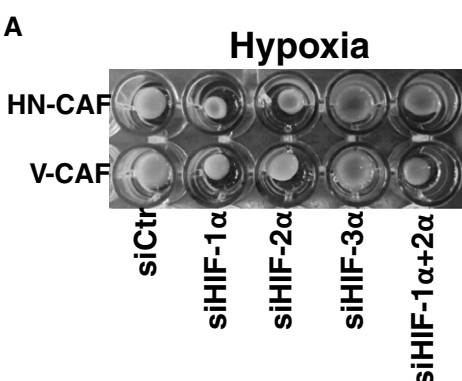

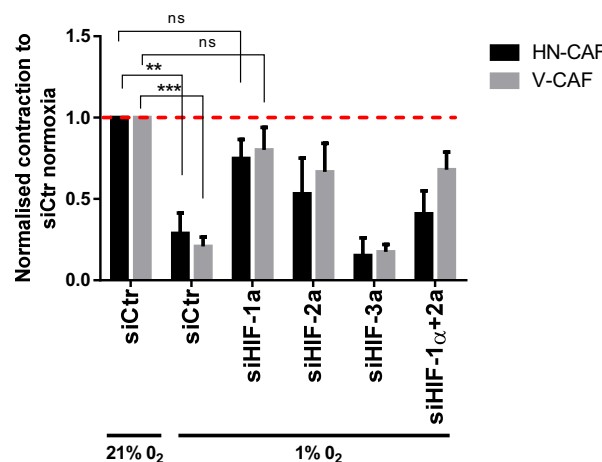

**Contraction assay in hypoxia**

**B**

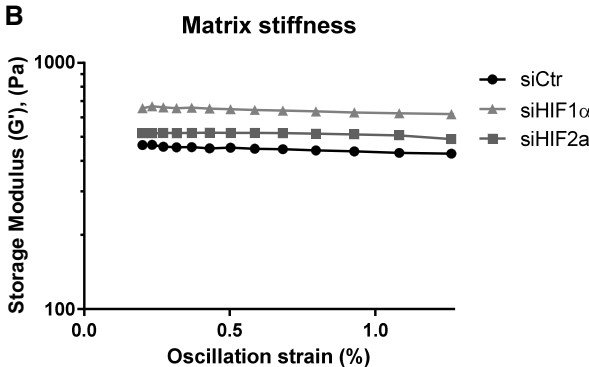

**C**

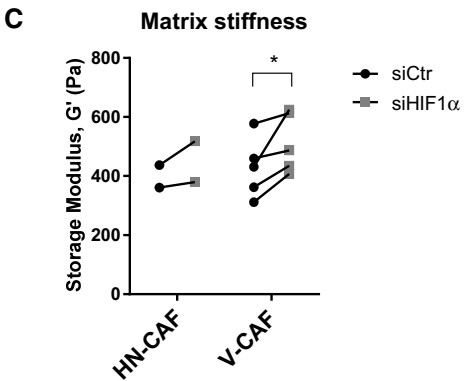

**D**

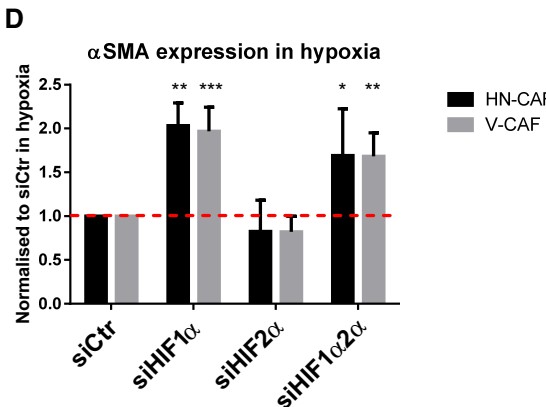

**E**

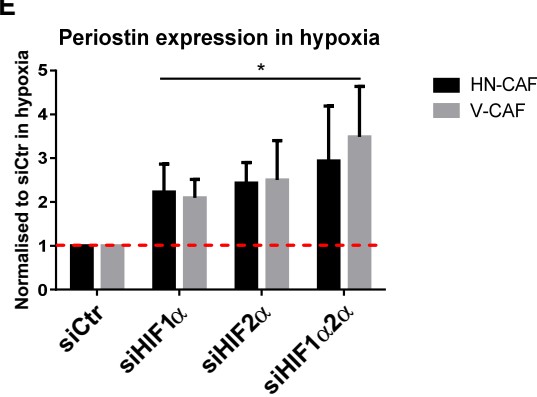

**Figure 4.  HIF-1α supports CAF-induced matrix remodelling and invasion.**

A    SiRNA-depleted CAFs were plated within collagen I/Matrigels and allowed to contract for 72 h under hypoxic conditions. Left panels show images of contraction. Bars shows mean ± s.d. quantification of contraction normalised to siRNA control in normoxia. *n* > 3 experimental repeats. ns, non-significant; **P < 0.01; ***P < 0.001; unpaired Student's *t*-test (two-tailed).

B, C    Matrix stiffness upon depletion of HIFs. Shear rheology of collagen I/Matrigels was performed 96 h after siRNA transfection. (B) Storage modulus was always measured over a decade of oscillation strain from 0.2 to 2%. (C) Scatter plot shows storage modules of matched experiments (siCtr vs siHIF1a). Each couple of match data points represents an independent experiment. *P < 0.05; paired Student's *t*-test (two-tailed).

D, E    HIFs regulate αSMA and periostin mRNA levels. CAFs were siRNA-depleted for HIFs, and mRNA levels of αSMA (D) and periostin (E) were quantified by qPCR 72 h post-transfection. CAFs were plated on gels under hypoxic conditions. Bars shows mean ± s.d. *n* > 3 experimental repeats. *P < 0.05; **P < 0.01; ***P < 0.001; unpaired Student's *t*-test (two-tailed).

In addition to being a common marker of activated fibroblasts, αSMA can enhance the mechanical strength of contractile actin bundles and can be sufficient to enhance fibroblastic contractile force [29]. Periostin has been shown to induce myofibroblast differentiation and activation during wound repair [30]. Therefore, we tested whether either αSMA or periostin is required for CAF function. When αSMA was depleted in human CAFs, we observed a loss in contractile force as measured by contraction of collagen matrices (Fig 3F). Similarly, depletion of periostin in human CAFs abrogated their ability to contract collagen matrices (Fig 3F). The knock-down efficiencies of αSMA and periostin are shown in Fig EV2B.

These data suggest that αSMA expression in fibroblasts should decrease in tumour areas of hypoxia as compared to well-oxygenated areas. We tested this hypothesis by injecting the hypoxy-probe into mice bearing 4T1 mammary tumours and stained tumour sections for PDGFRα and αSMA. While αSMA stains the activated pool of fibroblasts, PDGFRα stains a broader spectrum of fibroblasts, including 85% of mammary fibroblasts [31], and its expression is not affected by hypoxia (Fig 3A). Immunofluorescence analyses of primary 4T1 mammary tumours demonstrated a homogenous distribution of PDGFRα-positive fibroblasts (Fig 3G), whilst it clearly demonstrates reduced levels of αSMA-positive cells within hypoxic regions of the tumour (Fig 3H). These *in vivo* observations support our *in vitro* findings and suggest that αSMA-positive CAFs are either excluded or deactivated under hypoxic stress within the breast tumour.

### HIF-1α supports CAF-induced matrix remodelling and invasion

The hypoxic response is mediated in part through the induction of hypoxia-inducible transcription factors (HIFs). Indeed, depletion of HIF-1α and partly HIF-2α was able to rescue the hypoxic effect on CAF-induced contraction (Fig 4A) and matrix stiffening (Fig 4B and C), suggesting that HIFs act as suppressors of CAF-mediated matrix remodelling. This was further validated using multiple siRNAs targeting HIF-1α (Fig EV2C). Depletion of HIF-1α not only rescued loss of contraction but also the loss of CAF-induced SCC invasion under hypoxic conditions (Fig EV2D, knock-down efficiencies shown in Fig EV2E). Hypoxic downregulation of αSMA and periostin was also dependent on HIF-1α, as depletion of HIF-1α under hypoxic conditions re-established higher levels of αSMA and periostin mRNA (Fig 4D and E). Interestingly, HIF-2α also played a role in the hypoxic suppression of periostin (Fig 4E). These data establish that activation of HIF-1α leads to deactivation of CAFs under hypoxic conditions. Further, they support the reduction in αSMA and periostin expression as key events, leading to the reduced activity of CAFs.

### Loss of PHD2 suppresses CAF-induced matrix remodelling and invasion

In order to further characterise the molecular mechanisms coupling reduced oxygen levels to HIF-1α and CAF deactivation, we investigated the prolyl hydroxylase domain-containing proteins (PHD1–3). These are oxygen-dependent enzymes that target the alpha-subunit of HIF complexes for degradation under normoxic conditions [11]. Depletion of the three individual PHDs affected CAFs to various degrees when compared to control-depleted HN-CAFs: PHD1 depletion did not show any difference (Fig 5A and B), while PHD3 depletion

moderately affected elongation and matrix stiffening (Fig 5A and B). On the other hand, depletion of PHD2 phenocopied the response to hypoxia (Fig 5A–D). First, loss of PHD2 increased the length of the CAFs when cultured in 3D collagen matrices (Figs 5A and EV3A and B). This was confirmed using multiple siRNAs targeting PHD2 (Fig EV3A, knock-down efficiencies shown in Fig EV3C). Secondly, depletion of PHD2 significantly reduced the proficiency of CAFs to stiffen their surrounding matrix when compared to control-depleted cells (Fig 5B). This was further validated using atomic force microscopy (Fig 5C). Thirdly, loss of PHD2 prevented CAF-induced cancer cell invasion (Figs 5D and EV3D); this effect was confirmed using multiple siRNAs targeting PHD2 (Fig EV3E). Finally, the loss of PHD2 was also shown to suppress αSMA expression, but not the expression of periostin (Figs 5E and EV3F). The lack of effect on periostin levels may reflect its more complex regulation that also involves HIF-2α (Fig 4E). Consistent with the role for HIF-1α in regulating both αSMA and periostin (Fig 4D and E), we identified that PHD2 depletion stabilises HIF-1α in human CAFs (Figs 5F and EV3G). Further, depletion of HIF-1α completely rescued the loss of PHD2 in CAF-mediated SCC invasion (Figs 5G and EV3H). Taken together, these data support a mechanism whereby hypoxia regulates CAF-induced matrix remodelling and cancer cell invasion through PHD2-HIF-1α-driven mechanism.

### Chemical inhibition of PHDs reduces tumour stiffness and spontaneous metastasis

Increased tissue stiffness has proven to stimulate both skin and breast cancer progression in mouse models [7,32]. We therefore hypothesised that manipulation of PHD molecules could have beneficial effect on cancer progression by reducing the tumour stiffness. We chose to focus on breast cancer, as this type of cancer contains very high levels (~80%) of αSMA-positive fibroblasts [33]. We took advantage of the 4T1 breast cancer mouse model, as these primary tumours also contain high numbers of αSMA-positive CAFs [34]. We implanted mouse 4T1 mammary carcinoma cells into the mammary fat pad of syngeneic BALB/c mice and started to treat the mice every second day with the pan-PHD inhibitor; DMOG, once tumours were palpable (after 1 week). After 3 weeks of DMOG treatment, the primary tumours were excised and shear rheology was immediately performed on the fresh tissue. Treatment with DMOG significantly decreased tissue stiffness of the primary tumour (Figs 6A and EV4A–C) without affecting the tumour size (Fig 6B). We observed reduced αSMA and periostin mRNA levels in DMOG-treated tumours (Fig 6C). Immunohistochemical analysis revealed lower intensity αSMA staining but equivalent number of αSMA-positive cells (Fig EV4D); this mirrors the *in vitro* observations in Fig 3. We did not detect any gross differences in the distribution of collagen I and collagen III, as determined by Picrosirius red staining (Fig EV4D).

It has previously been shown that collagen cross-linking and increased tissue stiffness cooperate to promote breast cancer invasion through force-dependent mechanisms [7]. Consistently, softening of the primary tumour tissue caused by DMOG treatment reduced spontaneous metastasis to lungs and liver (Fig 6D). These data demonstrate that inhibition of PHD family enzymes reduces tumour stiffness, the expression of the key stromal fibroblast activation marker, αSMA, and metastasis.

The administration of DMOG targets all cell types within the tumour as well as targeting all three members of the PHD family.

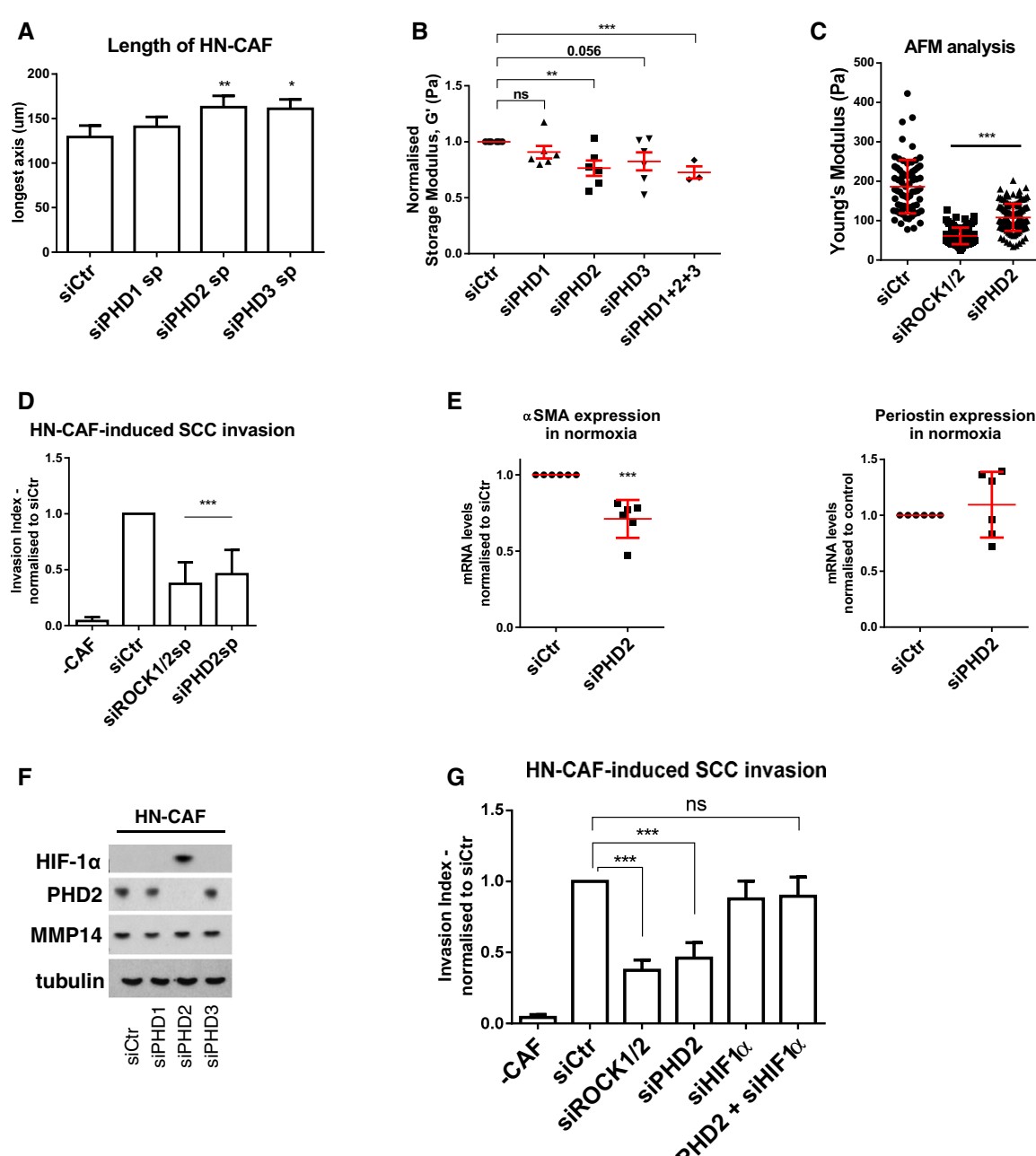

**Figure 5. PHD2 supports CAF-induced matrix remodelling and invasion.**

A   Depletion of PHD1–3 using smart pools of siRNAs regulates HN-CAF shape in 3D collagen I/Matrigels. Cells were fixed 72 h post-transfection and stained for F-actin. Bars show quantification of the longest distance of HN-CAFs in 3D matrices. Bars represent mean ± s.e.m. of four experiments. *P < 0.05; **P < 0.01; one-way ANOVA test.

B   Matrix stiffness upon depletion of PHD1–3. Shear rheology of collagen I/Matrigels was performed after 72 h of CAF remodelling. Scatter plot shows storage modules of matrices normalised to control siRNA-transfected HN-CAFs. Each data point represents an independent experiment. Line and error bars indicate mean ± s.e.m. ns, non-significant; **P < 0.01; ***P < 0.001; unpaired Student's t-test (two-tailed).

C   Atomic force microscopy of siRNA-depleted HN-CAFs. Scatter plot shows Young's modulus of matrices remodelled by HN-CAFs. Each data point represents a single measurement. Line and error bars indicate mean ± s.d. ***P < 0.001; one-way ANOVA test.

D   Loss of PHD2 in HN-CAF suppresses SCC12 cancer cell invasion. Bars represent mean ± s.d. n > 3 experimental repeats. ***P < 0.001; unpaired Student's t-test (two-tailed).

E   PHD2 regulates αSMA and periostin mRNA levels. HN-CAFs were siRNA-depleted for PHD2, and mRNA levels of αSMA and periostin were quantified by qPCR 72 h post-transfection. CAFs were plated on gels. n > 3 experimental repeats. ***P < 0.001; unpaired Student's t-test (two-tailed).

F   Immunoblotting analyses of the HIF-1α, PHD2 and MMP14. siRNA-depleted HN-CAFs were grown on gels for 72 h post-transfection.

G   HIF-1α depletion rescues the effect observed by PHD2 depletion in HN-CAFs-induced cancer cell invasion assay. Bars show mean ± s.d. HN-CAF-induced SCC12 cancer cell invasion normalised to siCtr. n > 3 experimental repeats. ns, non-significant; ***P < 0.001; unpaired Student's t-test (two-tailed).

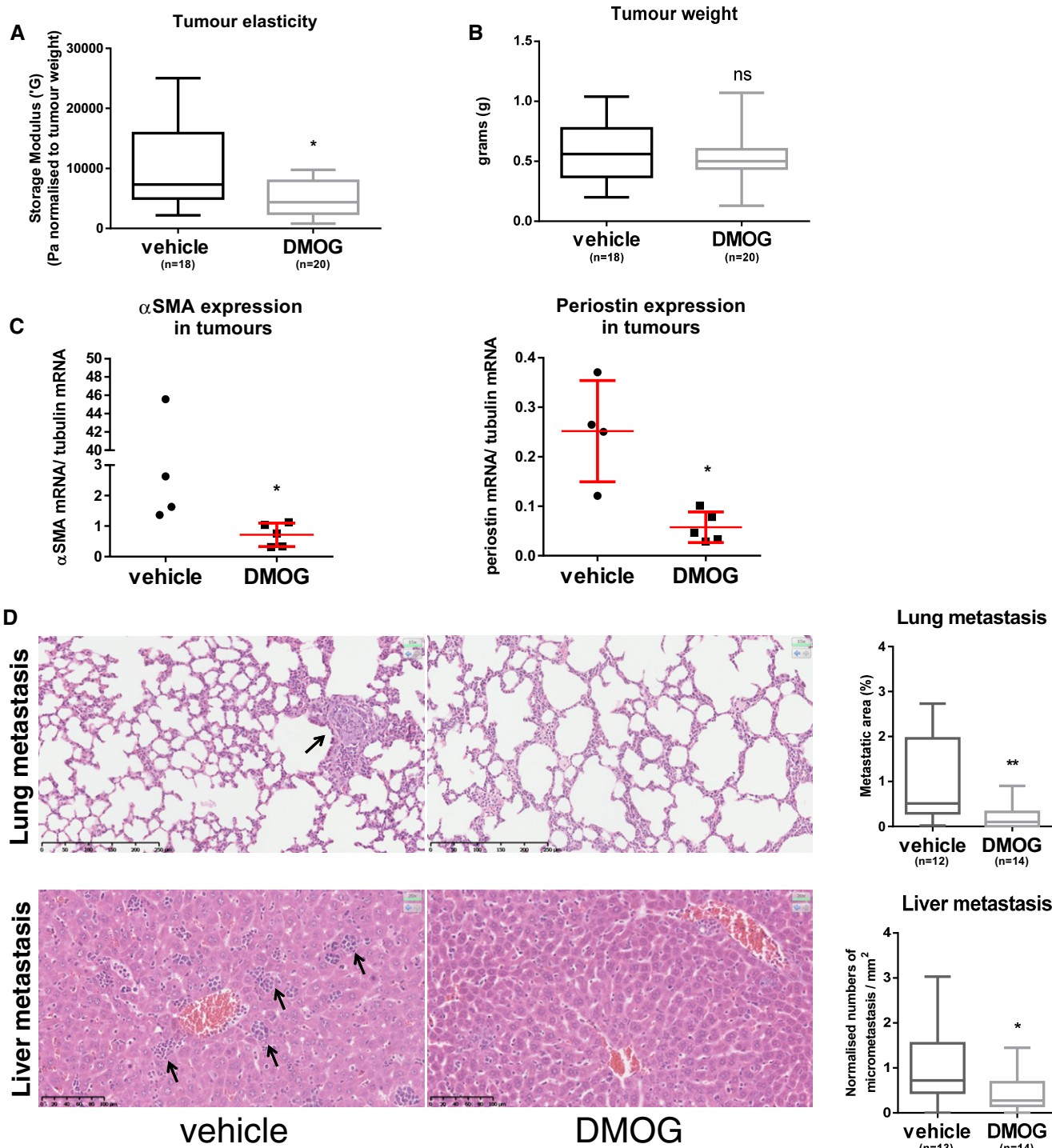

**Figure 6. Targeting PHD suppresses tumour stiffness and metastasis.**

A, B   Mice were orthotopically injected with 4T1 cells and treated with DMOG every second day. After 3 weeks, rheology was carried out on freshly excised primary tumour tissue. Four independent experiments were conducted. (A) Whisker plot shows storage modulus normalised to the total tumour weight of freshly excised primary breast tumour. (B) Whisker plot shows tumour weight. Number (*n*) of mice is indicated. Box and whiskers graph: line = median, box = distribution of 50% of values, whiskers = minimum to maximum. *$P < 0.05$; unpaired Student's *t*-test (two-tailed).

C   Periostin and αSMA mRNA levels are reduced in primary tumour samples after DMOG treatment. Periostin and αSMA mRNA levels were quantified by qPCR and normalised to tubulin mRNA levels. Each data point represents a single tumour. Line and error bars indicate mean ± s.d. *$P < 0.05$; unpaired Student's *t*-test (two-tailed).

D   H&E sections of the lung and liver metastases. Arrows indicate metastases. Whisker plot shows the quantification of spontaneous metastases. Box and whiskers graph: line = median, box = distribution of 50% of values, whiskers = minimum to maximum. *$P < 0.05$; **$P < 0.01$; unpaired Student's *t*-test (two-tailed). Scale bars: lung (top), 250 μm; liver (bottom), 100 μm.

We therefore sought to determine which cells within the tumour might be responsible for the reduced metastasis in DMOG-treated mice and which PHD enzyme was critical. In monoculture experiments, DMOG treatment did not inhibit the invasion of 4T1 cells (Fig EV5A), suggesting that inhibition of PHD enzymes in cancer cells cannot explain the reduction in metastasis. Depletion of PHD2 in endothelial cells, myeloid and T cells has also been shown to affect

tumour angiogenesis and cancer. However, DMOG treatment of 4T1 tumour-bearing mice did not alter the area of endomucin-positive blood vessels (Fig EV4D). We next turned to investigate the role of CAFs: co-injection of 4T1 with V-CAFs/shControl into nude mice increased lung and liver metastases as compared to 4T1 cells alone (Fig 7B). To validate the specific role of PHD2 in the CAF population, we co-injected 4T1 tumour cells with V-CAF/shPHD2. The knock-down

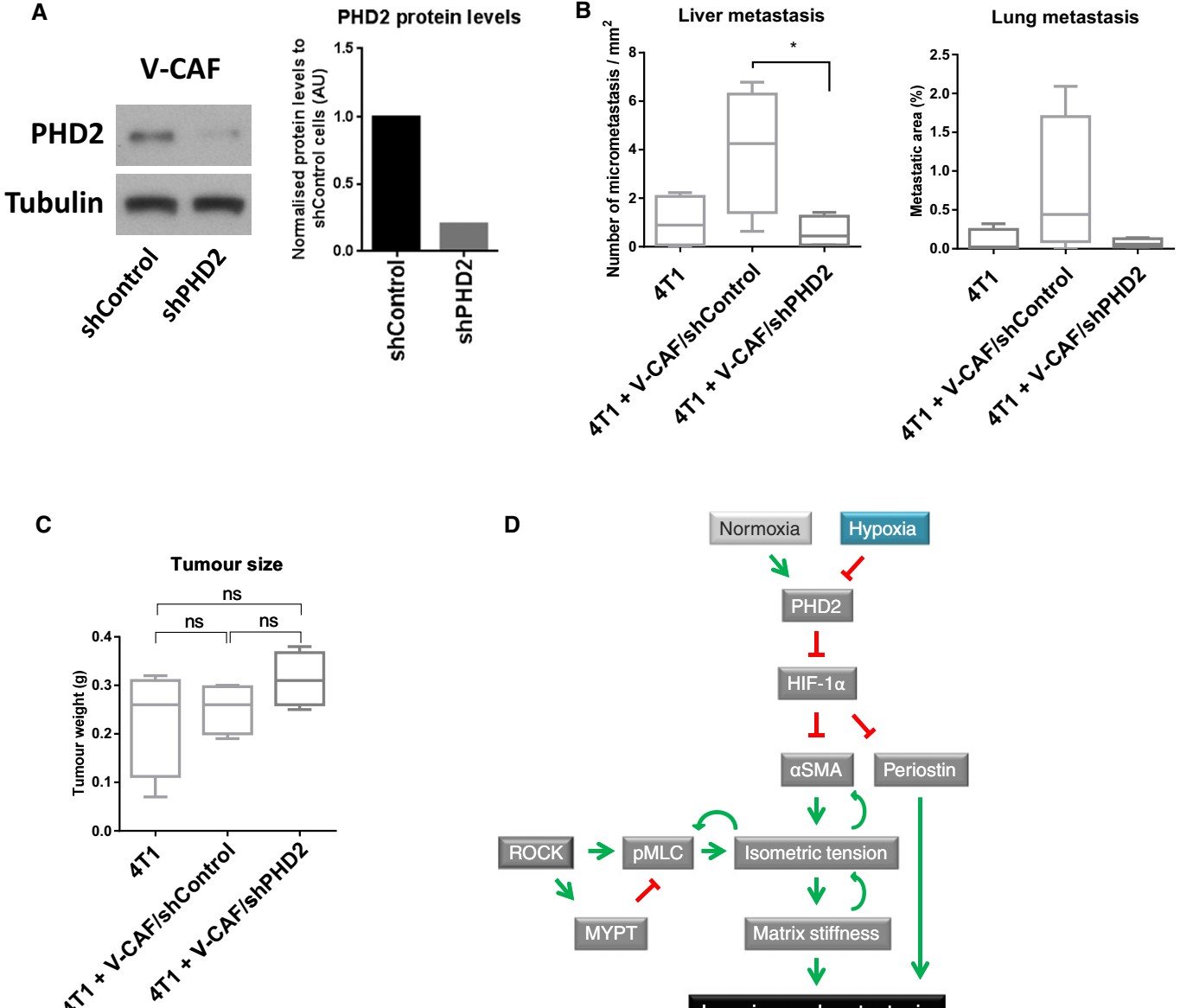

**Figure 7.  Loss of PHD2 in CAFs inhibits spontaneous metastasis.**

A    PHD2 knock-down efficiency of V-CAF/shPHD2 cells compared to V-CAF/shControl as demonstrated by Western blotting. Bars show quantification of PHD2 normalised to tubulin levels.

B, C    Mice were orthotopically co-injected with 4T1 cells and V-CAF/shControl, V-CAF/shPHD2. After 2 weeks, the tumours were taken out and processed. (B) Quantification of H&E sections of the lung and liver metastases. (C) Tumour size at the endpoint. Whisker plot shows the quantification of spontaneous metastases. Box and whiskers graph: line = median, box = distribution of 50% of values, whiskers = minimum to maximum. *n* = 4 mice per arm. ns, non-significant; *P < 0.05; unpaired Student's *t*-test (two-tailed).

D    Working model.

efficiency of PHD2 in V-CAFs using shRNAs is shown in Fig 7A. Co-injection of V-CAFs/shPHD2 prevented this additive effect of CAFs/shControl, returning the metastatic load to levels of 4T1 cells alone (Fig 7B). Of note, co-injection of CAFs in our model did not significantly alter primary tumour growth (Fig 7C).

These findings support the hypothesis that targeting PHD2 in CAF-enriched tumours, such as breast cancer, may have beneficial effects by decreasing spontaneous metastasis. In addition, these findings support the relationship between primary tumour stiffness and development of distant metastases observed in other mouse models [35,36].

## Discussion

The current thinking in cancer biology is that hypoxia and HIFs are generally considered to promote tumour cell aggressiveness, although there is some conflicting data. One possibility is that hypoxia has differing consequences on the different cell types present within tumours. Another possibility is that depending on the local surroundings, cells may react differently to hypoxia. Indeed, loss of PHD2 in myeloid and T cells inhibits tumour growth [19], while heterozygous loss of PHD2 in endothelial cells inhibits tumour metastasis through the upregulation of HIF-2α [18]. Furthermore, the loss of HIF-1α in stromal fibroblasts demonstrated accelerated tumorigenesis [17]. These studies prompted us to investigate the effect of low oxygen tension on CAFs.

Here, we show that prolonged exposure to hypoxia deactivates CAFs, ultimately leading to reduced remodelling and stiffening of their surrounding ECM. The ability of CAFs to remodel and invade their matrix was consequently impaired under hypoxia, and abrogated CAF-mediated invasion of cancer cells leading to decreased metastasis to distant organs (see model in Fig 7D).

The hypoxic events underlying CAF deactivation require the regulation of PHD2 and are dependent on levels of HIF-1α expression. Interestingly, the deactivation of CAFs does not occur immediately, but is a process which takes several days, as shown by the slow drop in pMLC and αSMA expression. These findings suggest that it is not merely an immediate transcriptional response by HIF, but rather an indirect and more complex regulation. Indeed, the minor effects of PHD3 and HIF-2α (Figs 4 and 5) may contribute to the deactivation of CAFs [37,38]. Heat-shock factor 1 (HSF1) and vitamin D receptor (VDR) have also been implicated CAF biology [39,40]; however, we did not observe any significant changes in HSF1 and VDR expression under hypoxia (Fig EV5C).

The slow reduction in isometric tension may be explained by the mutual regulation of matrix stiffness and CAF activation, as we have previously shown that CAF remodelling and matrix stiffness establish a self-reinforcing loop that drives further matrix stiffening and CAF activation [41]. It is therefore plausible that prolonged hypoxia and loss of PHD2 activity destabilise this positive feed-forward loop. This may be related to the low expression of periostin, an important inducer of both αSMA expression and myofibroblast activation [30,42–45]. Indeed, periostin knockout mice are protected from hyperoxia-induced alveolarisation and interstitial fibrosis due to the loss of αSMA-positive myofibroblasts within the lung tissue [42].

The fact that depletion of HIF-1α under hypoxic conditions rescues the loss of contraction and invasion suggests that HIF-1α

is a negative regulator of CAF-induced matrix remodelling. These data are consistent with the observation that low oxygen tension can reduce fibroblast function during wound healing [46]. In this context, skin myofibroblasts subjected to hypoxia de-differentiate and acquire a less contractile phenotype [46]. However, there may be variability in the response to hypoxia depending on other microenvironmental factors. We use soft substrates with physiologically relevant elastic modulus for all our functional assays, but fibroblasts may behave differently on rigid substrates. For example, on plastic or glass, hypoxia induces the expression of collagen and the collagen-remodelling enzymes P4HA1, P4HA2 and PLOD2 [47].

Our *in vitro* data suggested that the population of αSMA-positive fibroblasts could be expected to be lower in hypoxic areas of tumours. This was indeed what we observed in the fibroblast-rich 4T1 breast cancer model (Fig 3). Given that previous studies have successful eradicated subpopulations of CAFs and consequently improved the lifespan of tumour-bearing mice [48,49], we speculated whether deactivation of CAFs, by means of PHD inhibition, would have beneficial effects on tumour progression. Surprisingly, DMOG treatment reduced stiffening of the primary tumour and suppressed the metastatic burden to the lungs and liver (Fig 6). This was accompanied by a reduced αSMA and periostin expression within the primary tumour, thus suggesting that DMOG treatment suppresses CAF activation *in vitro* and *in vivo*. Most importantly, we were able to demonstrate that specific PHD2 depletion in the CAF population prevented CAF-induced metastasis to lungs and liver (Fig 7). Our data are consistent with studies showing that increased tumour stiffness promotes tumorigenesis in skin and breast [7,32]. Stromal HIF-1α can also modulate VEGF-A secretion and recruitment of immune cells [17]. The data also agree with data demonstrating the importance of fibroblast-produced periostin in spontaneous lung metastasis in the MMTV-PyMT mouse model [50].

Inhibition of oxygen sensors, that is the PHD molecules, has potential as a therapeutic for anaemia, ischaemic and inflammatory diseases [51,52]. However, only recently has interest grown in targeting of PHD molecules in cancer [18,53,54]. Several lines of evidence support targeting PHD2 in cancer [55–58], although some discrepancies have been observed [59]. Our data, in conjunction with others, support the notion that subtle changes in PHD and HIF activities are sufficient to reprogramme the tumour microenvironment. Indeed, treatment of DMOG every other day, as well as partial loss of PHD2 in V-CAFs, is sufficient to inhibit spontaneous metastasis in our models. Likewise, heterozygous deficiency of PHD2 in endothelial cells (presenting a 50% reduction compared to wild-type mice) is sufficient to restrain metastasis due to vessel normalisation [18]. These observations suggest the PHD2 manipulation is particular sensitive to the stromal compartment and a promising cancer target.

Our results highlight that globally targeting master regulators, such as PHD2, can have positive effects despite different tumour-promoting effects in different cell types when studied individually. In particular, we observe that targeting PHD2 in CAFs leads to their deactivation and reduces invasion and metastasis both *in vitro* and *in vivo*. Our study further supports PHD inhibition as a potential approach to prevent cancer progression through abrogating the metastasis-promoting effects of the tumour microenvironment.

# Materials and Methods

### Cell culture conditions

Human head and neck carcinoma-associated fibroblasts (HN-CAF) and human vulval carcinoma-associated fibroblasts (V-CAFs) were cultured as previously described [6,16]. All human CAFs were collected from fresh patient tissue according to informed consent, and immortalised using hTERT lentivirus. Characterisation of these human CAFs has previously been published [6,24,41]. Human SCC12, A431 and HT-1080 and mouse 4T1 cells were cultured as previously described [6,41,60]. All cell lines were regularly tested for mycoplasma and for murine pathogens by IMPACT testing (IDEXX Laboratories, Inc.). Cells were cultured at 37°C, 5% $CO_2$ and 21% (normoxic conditions) or 1% $O_2$ (hypoxic conditions) using a Whitley H30/H35 Hypoxystation (Don Whitley Scientific).

### CAF-induced cancer cell invasion assay

Organotypic culture system was set up as previously described [6]. Briefly, $5 \times 10^5$ HN-CAFs were embedded in a mixture of 4 mg/ml collagen I (#354249; BD Biosciences, Oxford, UK) and ~2 mg/ml Matrigel (#354234; BD Biosciences). After the gel was set at 37°C for 1 h, $5 \times 10^5$ SCC12, A431 or HT-1080 cells were plated on top in complete media for 24 h. The next day, the gel was mounted on a metal bridge and fed from underneath with complete media that were changed daily. After 5–7 days, the cultures were fixed using 4% paraformaldehyde plus 0.25% glutaraldehyde in PBS and processed by standard methods for haematoxylin and eosin (H&E) staining. For assays involving the removal of HN-CAFs, the fibroblasts were left to remodel the gel for 4 days, after which the gels were incubated in complete media plus puromycin (5 μg/ml) for 48 h to kill the fibroblasts. The gels were then washed three times with complete media (>30 min per wash) before overlaying them with $5 \times 10^5$ tumour cells. The assay then proceeded as mentioned above. The invasion index was calculated by measuring the total area over which cancer cells had dispersed (including invading and non-invading cells) and the area of non-invading cancer cells. The invasion index was then normalised to the invasion in normoxia- or to vehicle-treated samples.

### 4T1 spheroid invasion assay

For the 4T1 invasion assay, spheroids of $2 \times 10^4$ 4T1 cells were generated by hanging drop techniques in a volume of 20 μl including methylcellulose. The next day, the spheroids were embedded with collagen I/Matrigel as described above. The gels were let to polymerise for 1 h before overlaying with complete media. The spheroids were incubated in normoxia for 72 h with or without 1 mM DMOG. Phase contrast images were taken after 72 h of invasion. Images were binarised using ImageJ.

### Gel contraction assay

To assess force-mediated matrix contraction, CAFs (HN-CAFs: $5 \times 10^4$ cells and V-CAFs: $4 \times 10^4$ cells) were embedded in 100 μl of collagen I/Matrigel, yielding a final collagen concentration of approximately 4 mg/ml and a final Matrigel concentration of approximately 2 mg/ml, and seeded in 96-well plates. Once the gel was set (1 h), cells were washed once in normal media for 1 h and then replaced with fresh media with or without drugs added. Gel contraction was monitored after 72 h by taking photographs of the gels. To obtain the gel contraction value, the relative diameters of the well and the gel were measured using ImageJ software, and the percentage of contraction was calculated using the formula 100× (well diameter−gel diameter)/well diameter.

### Growth factors and inhibitors

The following drugs were used: Y27632, 10 μM (#1254; Tocris Bioscience); blebbistatin, 25 μM (#203391; Calbiochem/Merck); dimethyloxallyl glycine (DMOG), 1 mM (*in vitro*) and 8 mg/i.p. injection (*in vivo*), Cayman Chemical; $CoCl_2$, 1 mM (Sigma), *beta*-aminopropionitrile, 100–750 μM (Sigma); LOXL2 inhibitors (PXS1_1-5), 5 μM (Pharmaxis, Sydney, Australia).

### siRNA transfections

HN-CAFs and V-CAFs were transfected using Dharmafect 1 and 4 (Dharmacon), respectively. In brief, cells were plated at 60% confluence and subjected to transfection the following day using 50–100 nM siRNA. After 24 h of transfection, the cells were replated onto plastic or gels consisting of ~4 mg/ml collagen I (BD Biosciences; catalogue no. 354249) and ~2 mg/ml Matrigel (BD Biosciences; catalogue no. 354234). Cells were either fixed or lysed 48–72 h post-transfection. All siRNA oligos were purchased from Dharmacon and Qiagen. A comprehensive list of all siRNA oligos used in this study can be found in Table EV1.

### Generation of stable shRNA-CAFs

HN-CAFs and V-CAFs were infected with lentiviral constructs targeting PHD2. Generation and validation of the pHsH1-scramble-CMV-EGFP (shControl) and pHsH1-shHPH2-CMV-EGFP (shPHD2) constructs have already been performed [61]. The constructs were kindly provided by Dr. Kenneth Thirstrup, Lundbeck A/S. Generation of virus was done using second-generation packing system: the dr8.91 packaging vector and the VSV-G envelope vector. Viruses were produced in HEK293 cells. V-CAFs were infected with lentivirus and FACS sorted after selection. PHD2 knock-down efficiency was validated by Western blotting (Fig 7A).

### Western blotting, RNA extraction and quantitative RT–PCR

Western blotting was performed using standard techniques. A comprehensive list of all antibodies used in this study can be found in Table EV1. Total RNA was prepared using RNeasy (Qiagen), according to the manufacturer's instructions. cDNA was generated from DNase I-treated RNA (2 μg total RNA) using M-MLV H-point mutant reverse transcriptase (Promega) and random hexamer or Oligo dT primers. qPCR was carried out on cDNA using Lightcycler 480 SYBR green I Master Mix, according to the manufacturer's instructions. Several housekeeping genes were tested, and tubulin was chosen as the best control gene under both normoxic and hypoxic conditions. A comprehensive list of all qPCR primers used in this study can be found in Table EV1.

## Confocal microscopy and image analysis

All fluorescent images were acquired using a Zeiss LSM510 inverted confocal microscopes. For immunofluorescence analysis, the cells were fixed with 4% paraformaldehyde in PBS and permeabilised using 0.2% Triton X-100 in PBS before blocking with 3% bovine serum albumin in PBS. Cells were stained with DAPI and Phalloidin-TRITC (Sigma #P1951) for 2–6 h at RT. The shape and length of CAFs grown within 3D matrices were determined using Volocity software. Images were thresholded based on F-actin staining and cells identified in an automated manner.

## Flow cytometry

Expression of αSMA and pS19-MLC was analysed by flow cytometry after 72 h of hypoxic incubation. Cells were carefully detached and rinsed in PBS before fixation with 1% PFA/PBS for 1 h. Cells were permeabilised using 0.2% Triton X-100 in PBS before blocking with 3% bovine serum albumin in PBS. Cells were washed and stained with primary antibodies (see Table EV1) in FACS buffer (PBS, 1% FBS) for 2 h on ice. After extensive washing, the cells were incubated with secondary antibodies conjugated with Alexa488 for 2 h. Cells were washed and flow cytometry performed on a FACSCalibur (BD Biosciences).

## Histology and immunofluorescence of tissue sections

For immunohistochemistry (IHC), primary tumours and tissue samples were fixed in formalin, embedded in paraffin, sectioned and stained for H&E, alpha-smooth muscles actin (αSMA) and the hypoxyprobe-1, following standard histopathology operating protocols, as previously described [34]. All antibodies used can be found in Table EV1. Immunohistochemistry images were captured using NanoZoomer digital slide scanner (Hamamatsu). *Ex vivo* quantification of specific IHC stainings was carried out using Visiopharm, Denmark. For *ex vivo* quantification of tumour metastasis to liver and lung, scoring of H&E stained slides was carried out on the entire cross sections of the tissue ($n = 3$). For liver metastasis, the numbers of micrometastases (clusters of >4 tumour cells) were counted and normalised to the total area covered during quantification. For lung metastasis, the area covered by tumour cells was normalised to the total area covered during quantification. For immunofluorescence (IF), primary tumours were embedded in OCT, snap-frozen in −70°C isopentane and cryosectioned. Air-dried cryosections were briefly fixed in 4% PFA before being permeabilised with 1% Triton X-100 for 30 min, and subsequently blocked for 1 h with 5% serum from the same host as the secondary antibodies used, before 1 h incubation at RT with primary antibodies. After washing, sections were incubated with secondary fluorophore-conjugated antibodies for 1 h, washed and mounted with Mowiol with DAPI added to it. For multicolour labelling of markers, sequential labelling of antibody pairs was used. All washes were in TBS-T. IF images were captured using confocal microscopy (Zeiss LSM510 or LSM700).

## Shear rheology

Relative stiffness of CAF-remodelled collagen I/Matrigel matrices and primary mammary tumours were measured by shear rheology using a strain rotational rheometer (T.A. Instruments). In brief, CAFs were allowed to remodel 1 ml of collagen I/Matrigel matrices in 24-well dishes for 72 h prior to measurement. The primary tumour stiffness was measured on excised fresh tumours, 4 weeks after orthotopic mammary implantation as previously described [7]. Storage modulus was always measured over a decade of strain from 0.2% to 2% at a fixed angular frequency of 0.5 rad/s and a temperature of 21°C. The gels and tumours were found to be only minimally frequency dependent within the range of testing and showed a linear viscoelastic response within the strain range evaluated.

## Atomic force microscopy

Quantification of the elastic modulus of gels remodelled by CAFs was performed as previously described [41]. In brief, 24 h after siRNA transfection, $5 \times 10^4$ HN-CAFs were embedded in 100 μl of collagen I/Matrigel and seeded in 96-well plate. Once the gels were set, cells were maintained in CAF medium. After 3 days, the elastic modulus of the gels was measured.

## Animals and ethics statement

Female BALB/c immunocompetent and CD1 immunocompromised nude mice were purchased from Taconic. All experiments were carried out under authorisation and guidance from the Danish Inspectorate for Animal Experimentation. Female BALB/c mice (8 weeks old) were orthotopically injected into the mammary fat pad with $5 \times 10^5$ 4T1 cells and treated with DMOG (8 mg/injection) every second day. After 3 weeks of DMOG treatment, the mice were injected intraperitoneally with the hypoxyprobe™-1 (pimonidazole hydrochloride, hpi), 1 h prior to culling. Female CD1 nude mice (8 weeks old) were orthotopically injected into the mammary fat pad with a mixture of $1 \times 10^5$ 4T1 cells and $1 \times 10^6$ V-CAFs.

## Statistical analyses

All statistical tests were performed using one-way ANOVA, Sidak's multiple comparison test or paired/unpaired Student's *t*-test (two-tailed), *$P < 0.05$, **$P < 0.01$, ***$P < 0.001$.

Expanded View for this article is available online:
http://embor.embopress.org

## Acknowledgements

We acknowledge the support of FEBS long-term Fellowship for funding Chris D. Madsen and the Wellcome Trust-MIT Postdoctoral Fellowship for funding Emad Moeendarbary. This work was further supported by Cancer Research UK grant CRUK_A5317, the Francis Crick Institute (CDM and ES), the Novo Nordisk Foundation, Denmark, with a Hallas Møller Stipend (CDM and JTE), the Danish Cancer Society (R117-A7294-B2731) (JTP), the Danish Council for Independent Research YDUN grant (1084181001) (FVA) and the Innovation Fund Denmark (1311-00010B) (TRC). We thank Pharmaxis for supply of the LOXL2 inhibitors.

## Author contributions

CDM carried out all the experiments except those noted otherwise. JTP assisted with immunoblotting, qPCR and animal handling. LBS performed H&E stainings. FAV stained primary tumours for αSMA and hypoxyprobe. EM performed the atomic force microscopy under guidance of GC. TRC established

and set up experimental approaches for shear rheology and assisted with orthotopic breast cancer models. CDM, with the help of ES and JTE, designed the experiments and wrote the manuscript.

## Conflict of interest

The authors declare that they have no conflict of interest.

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
