## [Review Process File · EMBO Reports]

Manuscript EMBOR-2015-40107

Hypoxia and loss of PHD2 inactivates stromal fibroblasts to decrease tumour stiffness and metastasis

Chris D Madsen, Jesper T Pedersen, Freja A Venning, Lukram Babloo Singh, Emad Moeendarbary, Guillaume Charras, Thomas R Cox, Erik Sahai and Janine T Erler

Corresponding author: Janine T Erler, Erik Sahai, Chris D Madsen

Review timeline:

Submission date:	16 January 2015
Editorial Decision:	04 February 2015
Revision received:	17 July 2015
Editorial Decision:	30 July 2015
Revision received:	04 August 2015
Accepted:	04 August 2015

Transaction Report:

Editor: Nonia Pariente

1st Editorial Decision

04 February 2015

Thank you for your submission to EMBO reports. We have now received reports from the three referees that were asked to evaluate your study, which can be found at the end of this email. As you will see, although the referees find the topic of interest, two of them have concerns about the conceptual novelty of the work and all raise numerous concerns regarding the conclusiveness and completeness of the analyses.

Given the interest of the study, we would be happy to invite its revision. As the reports are below, I will not detail them here. However, it is clear that substantial additional experimentation to bolster causality, specificity, and provide a sufficient advance would be needed, as well as the inclusion of appropriate controls and quantifications throughout the study. All of the issues raised by the referees seem pertinent and should be addressed.

Please note that it is our policy to undergo one round of revision only and thus, acceptance of your study will depend on the outcome of the next, final round of peer-review. I appreciate that experimentally addressing all the referee concerns would involve extensive additional work of uncertain outcome and I would therefore be willing to reasonably extend our usual three months for revision, if it becomes necessary.

Please contact me if I can be of any assistance during the revision process.

REFEREE REPORTS:

Referee #1:

Summary

In this paper, the authors describe how cancer-associated fibroblasts (CAFs) are capable of sensing oxygen level via prolyl hydroxylase domain-containing protein 2 (PHD2, encoded by the *Egln1* gene). The authors demonstrate that CAFs' response to hypoxia leads to the reversion of the activated fibroblast phenotype (decreased α SMA, pMLC and pFAK levels) and to a decreased ability to remodel collagen I and to secrete extracellular matrix proteins (periostin, tenascin-C). Moreover, using an organotypic culture system, they show that, whereas in normoxic conditions CAFs were able to stimulate the invasion of tumour cells, CAFs lost their ability to do so in hypoxic condition and this was rescued by knocking-down HIF1 α in fibroblasts. Finally, using an in vivo model, the authors demonstrate that the inhibition of PHDs (systemic DMOG treatment) decreased primary tumour elasticity and metastasis formation.

This is an interesting study that focuses on a key component of the tumour microenvironment, the cancer-associated fibroblasts, and that identifies an important and novel regulatory mechanism of cancer-associated-fibroblast phenotype by hypoxia. The results of the PHD2 targeting experiment in tumours showing decreased metastasis are somewhat less novel but come in support of many studies (including on breast cancer cell lines) that have already shown that targeting PHD2/*Egln1* in tumour cells themselves (knockdown experiments) or systemic treatment with DMOG leads to smaller and/or less aggressive tumours. However, this report adds to the previous studies by demonstrating that tumours grown in DMOG-treated mice are less elastic.

The experimental design of the study is overall sound but the manuscript could be improved by providing a more thorough analysis and discussion of the results (see Specific Comments below). Finally, I believe this paper could appeal to a broad readership, as hypoxia and fibroblast activation are, beyond cancer, important for many other physiological and pathological processes.

Major comments:

The experimental design of the study is sound and the overall quality of the experiments is satisfactory however, some important controls are missing. In addition, most experiments could have been more thoroughly analyzed and the somewhat negative/contradicting results presented in the figures should be discussed.

1- The different CAF lines seem to behave differently and this should be commented in the text. From Fig 1A, it is not clear whether V-CAFs become more elongated when cultured in hypoxia, the authors should provide quantification and statistical analysis. Also, it seems that HN-CAFs but not V-CAFs are responsive to CoCl₂ (Fig 1C and 1D), can the authors comment on that. As V-CAFs seems to be less sensitive, why were they used and not HN-CAFs (more robust) in Fig S2A to monitor the effect of PHD2 knockdown on cell shape? Also, could the authors comment on the ability of V-CAFs to promote tumour cell invasion.

2- Western blot analysis Fig 1G and 1H:

- To be able to conclude on the elevation of the phosphorylation level of the proteins studied, it is important to include as controls the total fraction of MYPT, MLC, FAK and ERM proteins in addition to the actin which serves as a loading control. Even in absence of these controls, is the pool of phosphorylated ERMs really elevated under hypoxic condition or upon CoCl₂ or DMOG treatment? Similarly, FAK and MYPT decreased phosphorylation seems pretty modest, could the authors quantify the magnitude of the decrease?

- Do the authors detect a similar increase in HIF1 α expression in CAFs placed in hypoxia as the one seen upon DMOG treatment (PHD inhibition) or PHD2 knockdown?

3- Regarding gene expression studies:

- When possible, could the authors provide in addition to qPCR data, western blot analysis (Fig 4A, 4E, 5C, etc).
- For most of the knockdown experiments (Fig S2D, S3B), the authors only present qPCR data to show knockdown efficiency. As antibodies are available and used elsewhere in the manuscript, the authors, should provide data showing the decrease at the protein level by western blot. Also, as PHD2 and HIF1 are members of families of proteins, it would be interesting to show that upon knockdown the expression of the other members of the families is not affected.

4- Regarding the monitoring of the de-activation of CAFs under hypoxic condition:

- The results of the western blots presented in 4B and 4D seem different: 4B shows a decrease at 48hrs in the level of α SMA in HN-CAFs, but the decrease in 4D is observed at 72 and not 48 hours. How variable is the phenotype?
- The observation that, CAFs switch to a non-activated phenotype (α SMA^{low}/pMLC^{low}) and are not replaced by a different population is interesting. Could the authors provide immunofluorescence showing the expression of α SMA in CAFs cultured in vitro under hypoxia or normoxia?

5- In vivo tumour growth and metastasis assay: In this experiment the authors evaluate the effect of systemic treatment with DMOG on mammary tumour growth and metastasis.

- It would be interesting to know whether the decreased in α SMA and periostin expression in tumours is due to a decrease in the total number of CAFs in tumours growing in mice treated with DMOG. Could the authors perform immunohistochemical staining for these markers and FAP for example to evaluate that.

It would also be interesting to evaluate the level of HIF1 α in tumours treated or not with DMOG. Could the authors provide immunohistochemical staining of HIF1 α in the tumours? Do the authors observed more blood vessels in tumours grown in mice treated with DMOG?

- The results of PHD2/Egln1 targeting in tumours is not novel but comes in support of several studies that have shown that targeting PHD2/Egln1 in tumour cells (likely through a HIF1 α -independent mechanism) leads to smaller and/or less metastatic tumours (the authors should cite the appropriate references). However, some studies have also reported opposite results. In addition, hypoxic tumours are often (thought to be) more metastatic. The reason invoked for this is the increased angiogenesis in response to HIF-mediated up-regulation of VEGF. The manuscript would benefit from the addition of a short paragraph discussing the results of this report in the context of the broader literature and not only the publications that "coincide" with their findings.
- The demonstration that decreased elasticity correlates with decreased metastasis is an important finding. I would, however, suggest tuning down the last sentence of the paper as only correlative evidence between elasticity and metastasis is provided.

For those of us who only have a rudimentary knowledge of biomechanics: can the authors describe how elasticity relates to stiffness? Less elastic does not necessarily mean less stiff, or does it? It would also be interesting to evaluate how the elasticity correlates with the ECM content and organization of the tumours. Could the authors provide ECM staining of the tumours treated or not with DMOG (Masson's trichrome or Picrosirius Red staining). Although this may be beyond the scope of this report, have the authors tried to analyze the organization of collagen fibers using second harmonic microscopy?

6- Fig 5E: The scheme is a little bit confusing as, if read linearly, one could understand that hypoxia inhibits PHD2 which in turn inhibits HIF1 α . Whereas hypoxia (or PHD2 inhibition for that matter) leads to a stabilization of HIF1 α (due to a decrease in PHD2-mediated hydroxylation of HIF1 α leading to HIF1 α degradation).

Minor comments:

1- Several experiments are presented without statistical analysis (Fig S1C, S1D, S1E, S2B, etc.). Could the authors make sure to include the number of technical and biological replicates for in vitro experiments and provide appropriate quantitation and statistical analysis.

2- The experiment presented in Fig S1A is not clear. Could the authors improve rephrase the description of this experiment in the text. See page 9: "This prompted us to test CAFs on hard surfaces versus soft gels of physiologically relevant stiffness. These experiments did not demonstrate the same increase in collagens and related enzymes, when plated on gels as compared

to hard surfaces (Suppl. Fig. 1A)". What was exactly tested? What does "to test CAFs mean"? How do these results inform the main hypothesis?

3- The spelling of recurrent terms should be verified and homogenized through the manuscript. For example: collagen I (correct spelling) vs collagen-1 or collagen-I or references to the figures: "Fig" vs "Fig." vs "fig", etc.

Referee #2:

This is an ambitious and well-written study analyzing the impact of hypoxia, and the hypoxia sensors PHD and HIF, on the phenotypes of cancer-associated fibroblasts (CAFs).

Initial studies demonstrate that hypoxia alters CAF morphology and reduces the ability of CAFs to remodel matrix as determined by contraction assays and shear rheology (Fig. 1A-F). This occurs in a manner involving alterations in contractile regulators such as pMLC and pMYPT, without affecting CAF proliferation (Figs. 1G-I). It is subsequently demonstrated that a collagen-gels "primed" by CAFs under hypoxic conditions is less permissive for cancer cell invasion than gels "primed" by normoxic CAFs (Fig. 2). A series of siRNA experiments are suggesting that the hypoxia-mediated regulation of CAF-induced matrix remodeling and cancer cell invasion is controlled by PHDs and HIF1-alpha (Fig. 3). Furthermore, hypoxia is shown to alter gene-expression of CAFs including down-regulation of ASMA, which is shown in both tissue culture and experimental models (Fig. 4). Finally, treatment of experimental tumors with PHD inhibitors is shown to reduce lung and liver metastasis in a manner also involving reduced expression of ASMA and periostin in CAFs of primary tumors (Fig. 5).

In general these findings support previous literature (Kim, *Can Res*, 2013) on "CAF-deactivating" effects of hypoxia and add some novel mechanistic insight. At present stage it remains unclear if the study has the conceptual novelty associated with EMBO reports publications.

Major issues:

1. The initial phenotypic profiling of hypoxia-effects on CAFs include as endpoints CAF length, CAF contractility, matrix rheology, matrix stiffness, CAF proliferation and CAF-supported cancer cell invasion (Fig. 1 and 2). The siRNA studies analyzing the roles of HIF1 report on selected endpoints (Fig. 3). This should be corrected so that analyses with siHIF1 CAFs also include effects on matrix stiffness and CAF proliferation.
2. The animal experiment should be supplemented with some studies where the effects of PHD depletion or inhibition of CAFs is specifically analyzed. This could be done e.g. by studies with co-injection studies using control or PHD-depleted fibroblasts and subsequent analyses of DMOG effects.
3. CAF phenotypes are generally believed to also be controlled by cancer cell-derived signals. The study would therefore be more significant if it also included some analyses on how hypoxia affects cancer cell-induced "education" of fibroblasts.
4. The finding that periostin depletion reduces CAF contractility (Fig. 4G) suggest further analyses on potential periostin-dependency of the ability of CAFs to "prime" gels for cancer cell invasion.
5. The study, in its title and at other places (e.g. discussion second paragraph page 14), implies that PHD blockade in CAFs impairs metastasis. This is not supported by experiments since the animal experiment also includes PHD blockade in cancer cells.

Minor issues:

1. Animal studies should be supplemented by evidence that DMOG indeed blocks PHD in the experimental tumors.
2. The analyses of Fig. 4 H should be supplemented by analyses of some other fibroblast marker to indicate if the absence of ASMA signal in hypoxic region reflects changes in CAF phenotype or CAF abundance
3. The discussion about de-activation of CAFs (page 11) should refer to recent studies showing de-activation of CAFs after treatment with vitamin D3 or HSF1 down-regulation (Scherz-Shouval R,

Cell, 2014; Sherman, Cell, 2014)

Referee #3:

IN this study, Madsen et al investigate cancer associated fibroblasts are affected by loss of PHD2 and hypoxia in the context of tumours. They find that quite prolonged hypoxia leads to inactivation of these fibroblasts which leads to reduced tumour stiffness and reduced spontaneous metastasis. Overall this is a very interesting piece of work, raising questions as to whether PHD inhibition could be used as cancer therapeutics in the future. However, a number of important controls are missing which do not allow for all the conclusions to be supported by the data presented. Especially, whether it is only PHD2 that produces this effect, if the hypoxia effect can be seen earlier and whether the HIF-1 dependency is based on canonical activity even at this stage of the hypoxia response. Specific comments are below.

First of all, why was 72 hours of hypoxia chosen? At this stage cells are more or less adapted to hypoxia, was the phenotype seen earlier? Also, by the stage HIF-1alpha is mostly inactivated by increased levels of PHD2 and PHD3 in the feedback mechanism.

In figure 1I, how was proliferation measured. This is an important control that should be clearly investigated as previous reports have demonstrated that hypoxia has significant effects on proliferation. Again, a time course analysis should be provided here.

Figure 2 presents very clear results, however, it would be interesting to compare the results obtained in Figure 2C, with experiments where both CAFs and SCCs were exposed to hypoxia, a situation closer to reality.

Figure 3, depicts PHD2 dependency, however, PHD3 inhibition also has produced similar results. Also no statistics are presented for Figure 3G or 3I, these should be included. Rescue or gain of function experiments using PHD2 could be a good control to really demonstrate PHD2 dependency in this study. Also levels of PHD in the siRNA depletions are not provided, as these could explain the differences between the results obtained for the different PHDs (Figure 3H). Again important controls.

Figure 4 is missing stats in most graphs except 4F.

Figure 5, missing stats in graphs 5C. Assuming that mRNA is obtained from the tumours, how does the results correlate with HIF dependent targets such as CA9, VEGF or even PHD2. These are important controls since it would indicate that HIF-1 is still active in its canonical function or instead has an altered activity in this setting.

In the supplementary material, a number of figures have no stats on them: Sup Fig 1A, C, D, E, F. Sup. Figure 2B, does not have error bars or stats, no stats for Sup Fig 2C_D or Sup. Figure 3, with no stats on A, and no error bars or stats on B.

Response to reviewers' comments - EMBOR-2015-40107V1

General response

We were pleased that the reviewers found our study 'interesting' and that it would 'appeal to a broad readership' (reviewer #1), 'ambitious and well-written' (reviewer #2), and a 'very interesting piece of work' that could inform the feasibility of clinically targeting PHD enzymes (reviewer #3). We also thank the reviewers for their informed and diligent reading of our work and numerous suggestions for improving the study. In this revised manuscript we have significantly expanded the in vivo analysis and show the effects of targeting PHD2 specifically in CAFs, included more detailed kinetic analysis of the response to hypoxia, and added several experiments to further strengthen the work. A detailed point-by-point response to the reviewers' comments is provided below.

Referee #1:

Summary

In this paper, the authors describe how cancer-associated fibroblasts (CAFs) are capable of sensing oxygen level via prolyl hydroxylase domain-containing protein 2 (PHD2, encoded by the EglN1 gene). The authors demonstrate that CAFs' response to hypoxia leads to the reversion of the activated fibroblast phenotype (decreased α SMA, pMLC and pFAK levels) and to a decreased ability to remodel collagen I and to secrete extracellular matrix proteins (periostin, tenascin-C). Moreover, using an organotypic culture system, they show that, whereas in normoxic conditions CAFs were able to stimulate the invasion of tumour cells, CAFs lost their ability to do so in hypoxic condition and this was rescued by knocking-down HIF1 α in fibroblasts. Finally, using an in vivo model, the authors demonstrate that the inhibition of PHDs (systemic DMOG treatment) decreased primary tumour elasticity and metastasis formation.

This is an interesting study that focuses on a key component of the tumour microenvironment, the cancer-associated fibroblasts, and that identifies an important and novel regulatory mechanism of cancer-associated-fibroblast phenotype by hypoxia. The results of the PHD2 targeting experiment in tumours showing decreased metastasis are somewhat less novel but come in support of many studies (including on breast cancer cell lines) that have already shown that targeting PHD2/EglN1 in tumour cells themselves (knockdown experiments) or systemic treatment with DMOG leads to smaller and/or less aggressive tumours. However, this report adds to the previous studies by demonstrating that tumours grown in DMOG-treated mice are less elastic.

The experimental design of the study is overall sound but the manuscript could be improved by providing a more thorough analysis and discussion of the results (see Specific Comments below). Finally, I believe this paper could appeal to a broad readership, as hypoxia and fibroblast activation are, beyond cancer, important for many other physiological and pathological processes.

We thank the reviewer for noting that our study is 'interesting' and its broad appeal.

Major comments:

The experimental design of the study is sound and the overall quality of the experiments is satisfactory however, some important controls are missing. In addition, most experiments could have been more thoroughly analyzed and the somewhat negative/contradicting results presented in the figures should be discussed.

1- The different CAF lines seem to behave differently and this should be commented in the text. From Fig 1A, it is not clear whether V-CAFs become more elongated when cultured in hypoxia, the authors should provide quantification and statistical analysis. Also, it seems that HN-CAFs but not V-CAFs are responsive to CoCl₂ (Fig 1C and 1D), can the authors comment on that. As V-CAFs seems to be less sensitive, why were they used and not HN-CAFs (more robust) in Fig S2A to monitor the effect of PHD2 knockdown on cell shape? Also, could the authors comment on the ability of V-CAFs to promote tumour cell invasion.

We appreciate the opportunity to clarify this point. In fact, the V-CAFs are not less sensitive to hypoxic perturbations. Fig 1A now shows that V-CAFs do become elongated in hypoxia. Regarding the CoCl₂ experiment: we speculate there was a problem with one batch of the reagent in the V-CAF experiments originally presented. In subsequent experiments CoCl₂ did reduce collagen gel contraction by V-CAF. In light of these new data and the inconsistency with the original data we feel that the best thing is to take out the data on CoCl₂ in the contraction assays. The data using a different hypoxia mimetic that targets PHD enzymes, DMOG, is very consistent between HN-CAF and V-CAF (compare Figure 1C and Figure 1D). More generally, we would reiterate that we find a high degree of similarity in the response of HN-CAF and V-CAF to hypoxia (as measured by morphology, gel contraction, matrix rheology, α SMA, and periostin expression), DMOG (as measured by gel contraction, α SMA, and periostin expression), HIF1 α depletion (as measured by gel contraction, matrix rheology, and α SMA expression), and PHD2 depletion (as measured by morphology (Fig 5A&B and EV3A&B), gel contraction, and invasion). In particular, the new V-CAF tumour invasion data following PHD2 depletion are included in Figure EV3E. These data also confirm the ability of VCAF to promote cancer cell invasion.

2- Western blot analysis Fig 1G and 1H:

- To be able to conclude on the elevation of the phosphorylation level of the proteins studied, it is important to include as controls the total fraction of MYPT, MLC, FAK and ERM proteins in addition to the actin which serves as a loading control. Even in absence of these controls, is the pool of phosphorylated ERMs really elevated under hypoxic condition or upon CoCl₂ or DMOG treatment? Similarly, FAK and MYPT decreased phosphorylation seems pretty modest, could the authors quantify the magnitude of the decrease?

- Do the authors detect a similar increase in HIF1 α expression in CAFs placed in hypoxia as the one seen upon DMOG treatment (PHD inhibition) or PHD2 knockdown?

We thank the reviewer for raising these points. We have now excluded some of the western blot findings to focus on pMLC and pMYPT, and have provided quantifications (Fig 1). Regarding the normalisation of pMLC to total MLC levels, we agree that this would normally be the appropriate thing to do. However the situation with pMLC is complex; we have previously published that in CAFs actomyosin activity regulates YAP activity, and that, in turn, YAP is required for the expression of MLC (Figure 5D - Calvo et al Nature Cell Biology 2013, see below).

Figure 1. WB from figure 5d in Calvo et al, NCB, 2013

This generates a positive feedback loop and means that over long time periods the levels of pMLC and total MLC tend to co-vary (for example, see Figure 1D in Calvo et al Nature Cell Biology, see below).

Figure 2. WB from figure 1d in Calvo et al, NCB, 2013

This means that expressing the relative ratio of the two metrics can be misleading. Further, it is the total level of pMLC per cell that governs the contractile force generation. For these reasons, we feel that it is more appropriate to show quantification of pMLC normalised to the total number of cells (as approximated by actin or tubulin loading controls). We have also included western blot quantification of α SMA levels (Fig 3B).

We do observe an increase of HIF1 α in hypoxia but more modest compared to DMOG (see image below). We have not included this in the paper due to space limitations. PHD2 knockdown induces strong stabilisation of HIF1a (HN-CAF; Fig 5F and EV3H and V-CAF, EV3G). Overall, DMOG induces stronger stabilisation compared to hypoxia and PHD2 knockdown.

Figure 3. WB of HN-CAFs incubated 72h with hypoxia mimetic reagents or under hypoxia

3- Regarding gene expression studies:

- When possible, could the authors provide in addition to qPCR data, western blot analysis (Fig 4A, 4E, 5C, etc).
- For most of the knockdown experiments (Fig S2D, S3B), the authors only present qPCR data to show knockdown efficiency. As antibodies are available and used elsewhere in the manuscript, the authors, should provide data showing the decrease at the protein level by western blot. Also, as PHD2 and HIF1 are members of families of proteins, it would be interesting to show that upon knockdown the expression of the other members of the families is not affected.

We appreciate the reviewer's comment and have added western blot analysis to accompany qPCR data where possible throughout the manuscript (western blots of α SMA levels are included in Figures 3B & 3D, PHD2 knockdown is confirmed by western blot in Figures 5F, EV3, & 7A, α SMA and periostin knockdown is confirmed by western blot in Figure EV2B, and HIF1 α knockdown is confirmed by western blot in EV3H). Of note, the purpose of the qRT-PCR screen in Fig 3A was to screen for fibroblast markers. However, we have only included a handful of these markers in the paper (Fig 3A and Fig EV2). In fact we have extended this qRT-PCR screen to cover genes important for fibroblast biology (see figure below). In order to keep the paper focussed, we have only included western blot of targets of specific interest (α SMA and pMLC). If the reviewer feels we need to include the complete analysis in the main text, we will be happy to add it.

Figure 4. qPCR screen of various fibroblast markers. The data are normalised to normoxia. The data are from 48h of hypoxia.

The reviewer raises an interesting point regarding family members. We have experienced great difficulty in detecting HIF2 α , PHD1 and PHD3 by western blot. We therefore conducted an experiments looking at the mRNA changes over time in hypoxia, shown in the graph below. The data suggest that PHD2 is upregulated to compensate for the stabilised HIF1 α (this is supported by the western blot shown above). We are aware that this mRNA analysis will not pick up post-translational stabilisation of HIF2 α , but we hope the reviewer will understand that generation of reliable antibody reagents against HIF2 α to compensate for the deficiencies in the commercial reagents that we tested is outside the scope of revising a manuscript.

Figure 5F and EV3G also demonstrate that knockdown of PHD1 and PHD3 do not induce increased expression of PHD2.

Figure 5. mRNA analysis of PHD and HIF as a function of time in hypoxia

Following the reviewer’s suggestion, we have analysed the levels of PHD1&3 following PHD2 knockdown. We find that knockdown of PHD2 does induce an increase in PHD3 mRNA (see figure below). Nonetheless, this increase in PHD3 does not compensate for the reduced PHD2 levels in our functional assays. This leads us to conclude that PHD3 most likely plays a distinct role from PHD2 in the response of CAFs to hypoxia.

PHD mRNA expression after PHD2 depletion

Figure 6. mRNA analysis of PHDs after depletion of PHD2. Cells are plated on gels.

4- Regarding the monitoring of the de-activation of CAFs under hypoxic condition:

- The results of the western blots presented in 4B and 4D seem different: 4B shows a decrease at 48hrs in the level of α SMA in HN-CAFs, but the decrease in 4D is observed at 72 and not 48 hours. How variable is the phenotype?

- The observation that, CAFs switch to a non-activated phenotype (α SMA_{low}/pMLC_{low}) and are not

replaced by a different population is interesting. Could the authors provide immunofluorescence showing the expression of α SMA in CAFs cultured in vitro under hypoxia or normoxia?

We thank the reviewer for their observant points. The de-activation of α SMA occurs over several days. It is consistently reduced after 72 hours (Fig 3C); nonetheless, we agree with the reviewer that it is slightly variable after 48 hours. The slight differences observed may be due to the experimental preparation of the gels and batch-to-batch variation, as we have observed α SMA expression to decrease more rapidly if plated on softer matrices (see graph below). Thus, depending on the actual stiffness of the prepared matrix, this may slightly affect the expression of α SMA, thereby resulting in small variations in the readouts. Nonetheless, we would like to reiterate that decrease in α SMA protein is highly reproducible after 72hours of hypoxia.

Figure 7. mRNA levels of α SMA when cells are plated on plastic or 2 mg/ml collagen I gels for 72 hours

Regarding α SMA and pMLC staining: we have performed immunofluorescence of α SMA and pMLC in DMOG treated cells (shown below). In agreement with the western blot analysis, pS19-MLC is reduced. Further α SMA does not localise efficiently to stress fibres in DMOG treated cells. Currently, we have only performed this experiment once and therefore we are only showing it for the benefit of the reviewer. We are currently repeating these experiments. If the reviewer feels that these images should be included in the manuscript, then we would be happy to do so (assuming that the experiments underway corroborate the data shown below).

Figure 8. IF analysis of V-CAF treated with hypoxic mimetic agent (DMOG) and ROCK inhibitor (Y27632). Cells were stained for α SMA, pS19-MLC and F-actin.

5- *In vivo tumour growth and metastasis assay: In this experiment the authors evaluate the effect of systemic treatment with DMOG on mammary tumour growth and metastasis.*

All the comments and arguments stated under this point are very interesting. We have tried to address as many questions as possible with respect to the time allowed for the revision by the EMBO reports editor and remaining within the scope of the study, as detailed below (see EV4).

- It would be interesting to know whether the decreased in α SMA and periostin expression in tumours is due to a decrease in the total number of CAFs in tumours growing in mice treated with DMOG. Could the authors perform immunohistochemical staining for these markers and FAP for example to evaluate that.

In order to validate the total numbers of activated α SMA+ CAFs we quantified the total area covered by α SMA+ cells in mice treated with and without DMOG. The total area of α SMA staining is similar (Fig EV4D), however, the intensity is reduced in the DMOG treated tumours. This further supports our *in vitro* findings that α SMA levels are decreased in every cell, rather than eliminated in a subset (Figure 3E). We also stained tumours for PDGFR α , a well-known marker of normal fibroblasts that is not modulated by hypoxia (Fig 3A). PDGFR α was homogeneously expressed in the tissue in our model, also in the hypoxic areas (Fig 3G&H). This supports the notion that fibroblasts obviously can exist in hypoxic regions but that their activation state is decreased. Indeed, we observe a decrease in α SMA staining in hypoxic areas compare to normoxic areas (Fig 3H).

It would also be interesting to evaluate the level of HIF1 α in tumours treated or not with DMOG. Could the authors provide immunohistochemical staining of HIF1 α in the tumours? Do the authors observed more blood vessels in tumours grown in mice treated with DMOG?

HIF1 α staining in tissue is known to be very challenging and we were unable to get HIF1 α staining to work on our tissues. However, we can refer to the paper published by Taniguchi, C. M. *et al.* PHD inhibition mitigates and protects against radiation-induced gastrointestinal toxicity via HIF2. *Science translational medicine* **6**, 236ra264, doi:10.1126/scitranslmed.3008523 (2014). In this paper the authors demonstrate that DMOG treatment of mice using the same scheme as ours, stabilises HIF for up to 24 hours after DMOG administration. We inject every 2 days, meaning that our injection scheme is generating a fluctuating situation where HIF1 α is induced every 48 hours. In order to examine HIF1 α responses in our experiments we have stained for CAIX and endomucin (Fig EV4). We did not observe any obvious differences in CAIX, suggesting that the level of DMOG that we are administering does not trigger a full blown hypoxic response. We believe that there are two counter-acting events that explain this observation. The DMOG regime that we use does not cause a total blockade of PHD2 activity but causes a lower level modulation of PHD enzymatic activity that is probably similar to that in PHD2 \pm models (Leite d' Oliveira *et al* Cancer Cell). These show improved vascular perfusion that reduces hypoxia that would in fact lower CAIX expression. In agreement with this, we note that endothelial cell organisation (as judged by endomucin staining) is slightly altered in the DMOG treated tumours, (see image on page 22 of this rebuttal – DMOG treated tumours tend to have larger vessels with fewer sprouts, although the overall are of endomucin staining does not change). Although the overall endomucin staining does not change. These observations are in-line with those reported by Leite d'Oliviera *et al.* This possible reduction in CAIX is probably counter-balanced by a modest induction in CAIX resulting more directly from the action of DMOG on HIF1 α regulation in cancer cells. The stainings are derived from the same tumours upon which we performed the rheology measurements.

- The results of PHD2/Egln1 targeting in tumours is not novel but comes in support of several studies that have shown that targeting PHD2/Egln1 in tumour cells (likely through a HIF1 α -independent mechanism) leads to smaller and/or less metastatic tumours (the authors should cite the appropriate references). However, some studies have also reported opposite results. In addition, hypoxic tumours are often (thought to be) more metastatic. The reason invoked for this is the increased angiogenesis in response to HIF-mediated up-regulation of VEGF. The manuscript would benefit from the addition of a short paragraph discussing the results of this report in the context of the broader literature and not only the publications that "coincide" with their findings.

We apologise for not having included the papers describing PHD depletion in tumour cells. We have now added these, as well as incorporated into the text. As mentioned above, our endomucin staining could be consistent with some subtle modulation of angiogenesis. Nonetheless, we would respectively point out that the mechanism that we describe is very different the published work. We propose a key role for changes in fibroblast biology and matrix stiffness in the altered metastatic propensity of DMOG treated tumours.

- The demonstration that decreased elasticity correlates with decreased metastasis is an important finding. I would, however, suggest tuning down the last sentence of the paper as only correlative evidence between elasticity and metastasis is provided.

We have now tuned down our conclusions and statements to state that stiffness correlate with metastasis.

For those of us who only have a rudimentary knowledge of biomechanics: can the authors describe how elasticity relates to stiffness? Less elastic does not necessarily mean less stiff, or does it? It would also be interesting to evaluate how the elasticity correlates with the ECM content and organization of the tumours. Could the authors provide ECM staining of the tumours treated or not with DMOG (Masson's trichrome or Picrosirius Red staining). Although this may be beyond the scope of this report, have the authors tried to analyze the organization of collagen fibers using second harmonic microscopy?

With regard to the definitions of 'Elasticity and stiffness': The stiffness of a material is defined as the force applied to the material divided by the amount of the deformation. Therefore, the stiffness depends on the geometry (shape, size) of the material, but storage or elastic modulus (G') does not depend on geometry. For example, the storage modulus of an aluminium bar is the same as aluminium foil but their apparent "stiffness" to us is very different. Thus, in our rheological measurements, G' is directly comparable between gels since geometry is taken into consideration at time of measuring.

We can relate E' (Youngs modulus, or stiffness) directly to G' (the measured storage modulus) and Poisson's ratio (n') according to the following equation:

$$E' = G' \times 2(1+n')$$

E' = Young's Modulus

G' = Storage Modulus

n' = Poisson's ratio

For the purposes of our experiment we would assign a Poisson's ratio of 0.5 by assuming that our samples are homogenous, isotropic and in physics terms, incompressible. This is the value typically used for fibrillar matrices such as hydrogels (Anseth et al. 1996). Thus, we simplify our equation to: $\text{Stiffness} = 3xG'$

Therefore, assuming that two CAF containing collagen gels of similar geometry had differing G' , they would also exhibit a different 'stiffnesses'. The relationship in our case being that the value for stiffness (E') is approximately 3x that of the reported G'

We have now added quantification of picosirius red staining of the same tumours that we performed rheology on. The quantification shows that the area covered by collagen1&3 is not significantly decreased upon DMOG treatment (Fig EV4).

6- Fig 5E: The scheme is a little bit confusing as, if read linearly, one could understand that hypoxia inhibits PHD2 which in turn inhibits HIF1 α . Whereas hypoxia (or PHD2 inhibition for that matter) leads to a stabilization of HIF1 α (due to a decrease in PHD2-mediated hydroxylation of HIF1 α leading to HIF1 α degradation).

We thank the reviewer for highlighting this. We have made the scheme clearer.

Minor comments:

1- Several experiments are presented without statistical analysis (Fig S1C, S1D, S1E, S2B, etc.). Could the authors make sure to include the number of technical and biological replicates for in vitro experiments and provide appropriate quantitation and statistical analysis.

We thank the reviewer for pointing this out. We have updated the figure legends with number of repeats and have added p-values to the figures where appropriate. We have generated a table to facilitate the overview of the statistics (see below). Specifically, we have only added p-values when the manipulation is significant. However, in a few cases non-significance has been added to demonstrate exactly that.

Figure	Statistics
1A	unpaired student's t-test (two-tailed).
1C	one-way ANOVA test.
1D	paired student's t-test (two-tailed).
1F	paired student's t-test (two-tailed).
2B	unpaired student t-test (two-tailed).
2C	unpaired student t-test (two-tailed).
3A	unpaired student's t-test (two-tailed).
3B	unpaired student's t-test (two-tailed).
3C	unpaired student's t-test (two-tailed).
3E	unpaired student's t-test (two-tailed).
3F	unpaired student's t-test (two-tailed).
4A	unpaired student's t-test (two-tailed).
4C	paired student's t-test (two-tailed).
4D	paired student's t-test (two-tailed).
4E	unpaired student's t-test (two-tailed).
5A	one-way ANOVA test.
5B	unpaired student's t-test (two-tailed).
5C	one-way ANOVA test
5D	unpaired student t-test (two-tailed).
5E	unpaired student t-test (two-tailed).
5G	unpaired student t-test (two-tailed).
6A	unpaired student t-test (two-tailed).
6B	unpaired student t-test (two-tailed).
6C	unpaired student t-test (two-tailed).
6D	unpaired student t-test (two-tailed).
7B	unpaired student t-test (two-tailed).
7C	unpaired student t-test (two-tailed).
EV1B	unpaired student t-test (two-tailed).
EV1C	unpaired student t-test (two-tailed).
EV2A	unpaired student's t-test (two-tailed).
EV3A	one-way ANOVA test.
EV3B	one-way ANOVA test.
EV3E	unpaired student t-test (two-tailed).
EV4C	unpaired student t-test (two-tailed).
EV4D	unpaired student t-test (two-tailed).

2- The experiment presented in Fig S1A is not clear. Could the authors improve rephrase the description of this experiment in the text. See page 9: "This prompted us to test CAFs on hard surfaces versus soft gels of physiologically relevant stiffness. These experiments did not demonstrate the same increase in collagens and related enzymes, when plated on gels as compared to hard surfaces (Suppl. Fig. 1A)". What was exactly tested? What does "to test CAFs mean"? How do these results inform the main hypothesis?

We appreciate the reviewer's point. We have completely omitted this part of the manuscript, so as not to confuse the reader.

3- The spelling of recurrent terms should be verified and homogenized through the manuscript. For example: collagen I (correct spelling) vs collagen-1 or collagen-I or references to the figures: "Fig" vs "Fig." vs "fig", etc.

We thank the reviewer for highlighting these mistakes. We have now corrected these spellings

Referee #2:

This is an ambitious and well-written study analyzing the impact of hypoxia, and the hypoxia sensors PHD and HIF, on the phenotypes of cancer-associated fibroblasts (CAFs).

Initial studies demonstrate that hypoxia alters CAF morphology and reduces the ability of CAFs to remodel matrix as determined by contraction assays and shear rheology (Fig. 1A-F). This occurs in a manner involving alterations in contractile regulators such as pMLC and pMYPT, without affecting CAF proliferation (Figs. 1G-I). It is subsequently demonstrated that a collagen-gels "primed" by CAFs under hypoxic conditions is less permissive for cancer cell invasion than gels "primed" by normoxic CAFs (Fig. 2). A series of siRNA experiments are suggesting that the hypoxia-mediated regulation of CAF-induced matrix remodeling and cancer cell invasion is controlled by PHDs and HIF1-alpha (Fig. 3). Furthermore, hypoxia is shown to alter gene-expression of CAFs including down-regulation of ASMA, which is shown in both tissue culture and experimental models (Fig. 4). Finally, treatment of experimental tumors with PHD inhibitors is shown to reduce lung and liver metastasis in a manner also involving reduced expression of ASMA and periostin in CAFs of primary tumors (Fig. 5).

In general these findings support previous literature (Kim, Can Res, 2013) on "CAF-deactivating" effects of hypoxia and add some novel mechanistic insight. At present stage it remains unclear if the study has the conceptual novelty associated with EMBO reports publications.

We thank the reviewer for their helpful comments and the opportunity to improve our paper. We acknowledge the data from Kim et al. that suggest that HIF1 α depletion in FSP1+ stromal cells induces tumour growth; however, the mechanism they suggest is through the loss of VEGF and concomitant loss of tumour vasculature and infiltrating macrophages. In contrast, in this work the perturbations are not correlated to tumour growth, but instead to decreased contractile force generation within the fibroblast population. Loss of force generation in the fibroblasts population results in less tumour microenvironment ultimately leading to less metastasis. Our study also identifies the upstream regulator of HIF1 α (PHD2) and some of the downstream effectors such as α SMA and periostin. Importantly, we also identify that hypoxia within tumours may reduce activation of fibroblasts *in vivo*. A final point of difference is that our study tentatively suggests that not only HIF1 α is important but also that HIF2 α may be involved in the hypoxia-induced reversion of CAFs to a less activated state (Figure 4A, B, & E).

To summarise, our study may be viewed as complementary to the work of Kim et al but it is clearly distinct its molecular focus on PHD2 and the identification of downstream biophysical changes that facilitate metastasis. We believe that our identification of a PHD2 as a regulator of CAF activity and its potential as 'CAF deactivator' may provide us with new strategies to treat cancer patients. Indeed, many PHD drugs are currently being developed.

Major issues:

1. The initial phenotypic profiling of hypoxia-effects on CAFs include as endpoints CAF length, CAF

contractility, matrix rheology, matrix stiffness, CAF proliferation and CAF-supported cancer cell invasion (Fig. 1 and 2). The siRNA studies analyzing the roles of HIF1 report on selected endpoints (Fig. 3). This should be corrected so that analyses with siHIF1 CAFs also include effects on matrix stiffness and CAF proliferation.

We thank the reviewer for noting this. We have now included more analysis to confirm that HIF1 α is able to rescue the hypoxic effect. Specifically, we show that that siRNA targeting HIF1 α in hypoxia increases stiffening of the matrix (Fig 4B&C). Further, we demonstrate that siRNA against HIF1 α in hypoxia returns α SMA and periostin to the levels observed in normoxic cells (Fig 4D). We have not included any analysis of proliferation as we do not observe any changes in CAF proliferation under any hypoxic perturbations (Fig. EV1D).

2. The animal experiment should be supplemented with some studies where the effects of PHD depletion or inhibition of CAFs is specifically analyzed. This could be done e.g. by studies with co-injection studies using control or PHD-depleted fibroblasts and subsequent analyses of DMOG effects.

We agree with the reviewer and think this is an excellent suggestion. We have now performed experiments where we co-injected 4T1 cells with shControl and shPHD2 CAFs. Figure 7 shows that co-injection of V-CAF with 4T1 leads to increased metastasis. Importantly, we further show that shRNA depletion of PHD2 in the CAF population abrogates CAF-induced 4T1 metastasis to liver and lungs (Fig 7).

3. CAF phenotypes are generally believed to also be controlled by cancer cell-derived signals. The study would therefore be more significant if it also included some analyses on how hypoxia affects cancer cell-induced "education" of fibroblasts.

We agree with the reviewer that this would be very interesting to investigate. We have conducted some experiments to answer this question. The results are included below for the benefit of the reviewer. Briefly, we find that conditional media from hypoxic 4T1 cells does not greatly affect fibroblasts in our contraction assays nor does it modulate α SMA expression in fibroblasts (see figure below). Given the rather inconclusive nature of these experiments and the large amount of data already in the manuscript, we feel that including these investigations would detract from the focus and as they are not directly within the scope of our study.

Figure 9. Contraction assay using various conditional media from 4T1 cells cultured for 24 hours in hypoxia. The bars show quantification of the contraction assay.

Figure 10. aSMA expression of normal fibroblast cultured for 24 hours with conditioned media generated from 4T1 and TS2 cells in hypoxia for 24 hours.

4. The finding that periostin depletion reduces CAF contractility (Fig. 4G) suggest further analyses on potential periostin-dependency of the ability of CAFs to "prime" gels for cancer cell invasion.

We agree with the reviewer that this is very interesting. However, periostin expression in fibroblasts has already been shown to be important for metastasis (Malanchi et al Nature 2011); therefore, although a more detailed investigation of its role in invasion specifically would be interesting, we feel it would provide only an incremental advance. The key point that we wish to make is that

periostin expression is modulated by hypoxia in CAFs via a PHD2/HIF1 α /HIF2 α mechanism and that this regulation contributes to the de-activation of CAFs by hypoxia.

5. The study, in its title and at other places (e.g. discussion second paragraph page 14), implies that PHD blockade in CAFs impairs metastasis. This is not supported by experiments since the animal experiment also includes PHD blockade in cancer cells.

We acknowledge the reviewer's point. These statements are now justified with the inclusion of the new data showing that co-injection of CAF/shPHD2 prevents CAF-induced metastasis of 4T1 to lungs and liver (Fig. 7).

Minor issues:

1. Animal studies should be supplemented by evidence that DMOG indeed blocks PHD in the experimental tumors.

Our collaborators already published the effect of DMOG administration using the exact same protocol. For further information see Taniguchi, C. M. *et al.* PHD inhibition mitigates and protects against radiation-induced gastrointestinal toxicity via HIF2. *Science translational medicine* **6**, 236ra264, doi:10.1126/scitranslmed.3008523 (2014). In this paper the authors demonstrate that DMOG treatment of mice using the same scheme as ours, stabilises HIF for up to 24 hours after DMOG administration. We inject every 2 days, meaning that our injection scheme is generating a fluctuating situation where HIF1 α is induced every 48 hours.

2. The analyses of Fig. 4 H should be supplemented by analyses of some other fibroblast marker to indicate if the absence of ASMA signal in hypoxic region reflects changes in CAF phenotype or CAF abundance

This is a valid point. We have now added PDGFR α staining as this is a robust marker for resident fibroblasts in the mammary fat pad. Indeed, the staining suggests that these fibroblasts are homogeneously distributed in both normoxic and hypoxic areas (Fig3G&H).

3. The discussion about de-activation of CAFs (page 11) should refer to recent studies showing de-activation of CAFs after treatment with vitamin D3 or HSF1 down-regulation (Scherz-Shouval R, Cell, 2014; Sherman, Cell, 2014).

We thank the reviewer for highlighting these studies. We now include these papers in our discussion. In addition, we include data that suggest that VDR and HSF1 expression levels were not significantly altered in prolonged hypoxia in our model system (Fig EV5C).

Figure 11. mRNA levels after 48 hours on 2 mg/ml collagen I gels

Referee #3:

IN this study, Madsen et al investigate cancer associated fibroblasts are affected by loss of PHD2 and hypoxia in the context of tumours. They find that quite prolonged hypoxia leads to inactivation of these fibroblasts which leads to reduced tumour stiffness and reduced spontaneous metastasis.

Overall this is a very interesting piece of work, raising questions as to whether PHD inhibition could be used as cancer therapeutics in the future. However, a number of important controls are missing which do not allow for all the conclusions to be supported by the data presented. Especially, whether it is only PHD2 that produces this effect, if the hypoxia effect can be seen earlier and whether the HIF-1 dependency is based on canonical activity even at this stage of the hypoxia response.

Specific comments are below.

We thank the reviewer for noting that our study is very interesting and may have clinical implications.

First of all, why was 72 hours of hypoxia chosen? At this stage cells are more or less adapted to hypoxia, was the phenotype seen earlier? Also, by the stage HIF-1alpha is mostly inactivated by increased levels of PHD2 and PHD3 in the feedback mechanism.

We appreciate the reviewer's point. The reviewer is quite correct that hypoxic responses can occur rapidly and feedback mechanisms may diminish the magnitude of the effects at longer time points. In Figure 3 we have now evaluated different time points of hypoxia. α SMA mRNA expression begins to decline after 24 hours and becomes maximal between 48-72 hours. α SMA protein expression begins around 48 hours and is consistently reduced by 72 hours (see also reviewer #1 point#4). A further important aspect is that fibroblasts need time to remodel their surrounding microenvironment, both *vivo* and *in vitro*. The gel contraction, invasion, and metastasis assays all take several days, therefore we wished to study the state of fibroblasts in a timeframe that corresponds well with our functional assays. Our data clearly show that in CAFs HIF1 α levels are elevated at 72 hours (Fig 1G). As the reviewer correctly predicts, PHD2 levels are also increased at this time, but it is not enough to induce degradation of HIF1 α as shown in Fig 1G and below for the benefit of the reviewer. We also observe increased PHD1 mRNA in CAFs after 2-3 days, but HIF1 α mRNA is unchanged suggesting that the elevated HIF1 α proteins are the result of a post-translational mechanism (see figure below).

To conclude, the response to hypoxia clearly persists in CAFs at the longer time points that are relevant for our functional assays. Further, the importance of prolonged hypoxia is clinically relevant not only in cancer but also in various healing processes and chronic fibrosis.

Figure 12. (left) WB of HN-CAFs incubated 72h in hypoxia. (right) mRNA analysis of PHD and HIF as a function of time in hypoxia.

In figure 11, how was proliferation measured. This is an important control that should be clearly investigated as previous reports have demonstrated that hypoxia has significant effects on proliferation. Again, a time course analysis should be provided here.

We thank the reviewer for this point. We measured proliferation by counting cells every day both manually in adherent cultures and using automatic cell counting of trypsinized cells. No significant differences were observed in normoxia versus hypoxia when cells were on plastic or gels (Fig EV1D). We also cultured the CAFs for up to 3 weeks in hypoxia without observing any changes in proliferation (data not shown).

Figure 2 presents very clear results, however, it would be interesting to compare the results obtained in Figure 2C, with experiments where both CAFs and SCCs were exposed to hypoxia, a situation closer to reality.

Actually, Figure 2 does show data where both CAFs and SCCs have been cultured under hypoxia. Perhaps the reviewer is suggesting an experiment where we co-culture CAFs and cancer cells under normoxic and hypoxic conditions? We have performed this experiment but not included the data in the revised manuscript due to space limitations and in order to keep within the scope of the study. The results displayed in the graph below show that co-culturing CAFs and SCC12 cancer cells in hypoxia decreases SCC12 invasion as compared to normoxia.

Live co-culture invasion assay

Figure 13. Organotypic invasion assay where HN-CAFs and SCC12 are co-cultured together during the entire length of the experiment.

Figure 3, depicts PHD2 dependency, however, PHD3 inhibition also has produced similar results. Also no statistics are presented for Figure 3G or 3I, these should be included. Rescue or gain of function experiments using PHD2 could be a good control to really demonstrate PHD2 dependency in this study. Also levels of PHD in the siRNA depletions are not provided, as these could explain the differences between the results obtained for the different PHDs (Figure 3H). Again important controls.

We agree with the reviewer. The PHD3 knockdown is changing cell shape and is close to significant in the shear rheology experiment ($p=0.056$ Fig 5B). This is very interesting, and it is possible that PHD3 is cooperating functionally with PHD2 in the regulation of matrix remodelling and invasion. Indeed, co-knockdown of all three PHD1-3 decreased matrix stiffness better than PHD2 alone (Fig 5B). However, the greater magnitude and statistical significance of the data obtained with PHD2 siRNA led us to focus on PHD2 in this study. We have now included data showing that PHD2 specifically regulates HIF1 α (Fig 5F) and α SMA (Fig4D), while periostin is co-regulated by HIF1 α and HIF2 α (Fig 4E). It is plausible that PHD3/HIF2 α assist in the regulation of periostin, but we can only speculate at this moment.

Regarding the knockdown efficiencies and PHD2 re-expression: we now show PHD2 western blots to confirm depletion in Figure 5F & EV3. These blots also show that PHD2 depletion does not affect PHD1 or PHD3 protein levels. We show that 4 different PHD2 siRNA effectively knockdown the protein (EV3C) and that they have similar effects on CAF morphology and ability to promote invasion. These data with four independent siRNA make us very confident in the role of PHD2 in CAF biology and therefore we have not performed the re-expression of siRNA resistant PHD2. We have performed qRT-PCR demonstrating the efficacy of PHD1&3 depletion (Fig EV3C and figure below) – unfortunately our numerous attempts to get reliable PHD1&3 western blots were unsuccessful. Although it is always problematic to compare the efficiency of knockdown between different proteins when using western blots to evaluate one and qRT-PCR to evaluate the other, it seems

quite likely that the PHD3 depletion is less efficient. This could explain the more marginal effects we observe with PHD3 depletion. We have now moderated our language so as not to imply that PHD2 is the only relevant prolyl hydroxylase in CAFs. We did not go into further details due to time limitations and the scope of our study. However, these will form the basis of subsequent new studies.

Figure 14. Knockdown efficiencies of PHD1 and PHD3 by qPCR.

Figure 4 is missing stats in most graphs except 4F.

We apologise for this and have now added p-values to the figures.

Figure 5, missing stats in graphs 5C. Assuming that mRNA is obtained from the tumours, how does the results correlate with HIF dependent targets such as CA9, VEGF or even PHD2. These are important controls since it would indicate that HIF-1 is still active in its canonical function or instead has an altered activity in this setting.

Again, we apologise for this and have added p-values to the figures. We thank the reviewer for their suggestion to look at HIF targets, however, we would like to clarify that our DMOG administration protocol is identical to the one used by our collaborators who have already published the effect of DMOG administration using the exact same protocol. For further information see Taniguchi, C. M. *et al.* PHD inhibition mitigates and protects against radiation-induced gastrointestinal toxicity via HIF2. *Science translational medicine* 6, 236ra264, doi:10.1126/scitranslmed.3008523 (2014). In this paper the authors demonstrate that DMOG treatment of mice using the same scheme as ours, stabilises HIF for up to 24 hours after DMOG administration. We inject every 2 days, meaning that our

injection scheme is generating a fluctuating situation where HIF1 α is induced every 48 hours, and not a maximal inhibition of PHD enzymes.

In response to the reviewer's request to examine HIF1 α responses in our experiments, we have stained for CAIX and endomucin (Fig EV4 and shown below). We did not observe any obvious differences in CAIX, suggesting that the level of DMOG that we are administering does not trigger a full blown hypoxic response. We believe that there are two counter-acting events that explain this observation. The DMOG regime that we use does not cause a total blockade of PHD2 activity but causes a lower level modulation of PHD enzymatic activity that is probably similar to that in PHD2 \pm models (Leite d' Oliviera et al Cancer Cell). These show improved vascular perfusion that reduces hypoxia that would work in fact lower CAIX expression. In agreement with this, we note that endothelial cell organisation (as judged by endomucin staining) is slightly altered in the DMOG treated tumours (see image below – DMOG treated tumours tend to have larger vessels with fewer sprouts), although the overall are of endomucin staining does not change. These observations are in line with those reported by Leite d'Oliviera et al. This possible reduction in CAIX is counter-balanced by a modest induction in CAIX resulting more directly from the action of DMOG on HIF1 α regulation in cancer cells. It is important to note that the stainings are derived from the same tumours upon which we performed the rheology measurements.

CAIX

Figure 15. IHC staining of CAIX in breast tumor sections.

Endomucin

Figure 16. IHC staining of endomucin in breast tumor sections

In the supplementary material, a number of figures have no stats on them: Sup Fig 1A, C, D, E, F. Sup. Figure 2B, does not have error bars or stats, no stats for Sup Fig 2C_D or Sup. Figure 3, with no stats on A, and no error bars or stats on B.

We apologise for this and have now added p-values to the figures where significant differences are observed.

Thank you for the submission of your revised manuscript to our offices. We have now received the enclosed reports from the three initial referees. As you will see, they are now all supportive of publication, although referees 1 and 3 raise some minor issues that need to be taken care of. They all concern clarifications, except for the need to add some analysis of reproducibility (error bars) to some of the data, as requested by referee 3. We would not insist on adding error bars to figure 7A, but we do ask that you include them in EV1D and EV2C.

In addition, there are a few omissions regarding the mathematical analyses of the data in several figures. The number of experiments seems to be missing from figures 1G, 1H, 3C, EV1F and EV2D. In the case of EV2E, EV3C and EV3F, please ensure that all information on the number of independent experiments measured, the type of error bars used and statistical test applied to the data (if applicable), is included.

We will indeed include your study in full article format; thank you for reformatting it.

I look forward to seeing a new revised version of your manuscript as soon as possible.

REFEREE REPORTS:

Referee #1:

The revised version of the manuscript by Madsen et al. is very satisfactory. The experiment showing that the co-injection of 4T1 tumor cells with shPDH2 CAFs reduced the additive effect on metastasis formation observed when co-injecting tumor cells with CAFs as compared to tumor cells alone (Figure 7B) is a significant improvement. Pending very minor corrections (see list below), I believe that the revised version of the manuscript is suitable for publication in EMBO Reports.

Minor corrections:

- The list of authors has been changed since the initial submission. The authors should update accordingly the "Author contribution" section of the manuscript.
- The legend of Figure 3 should be relabeled: what is currently labeled G in the text likely corresponds to Figure 3F.
- Figure 7D is still confusing (misleading?), especially the inhibiting mark from PHD2 to HIF-1 α . Instead of the attempt at using a color code, could the authors indicate with up or down arrows the direction of the regulation? Maybe the scheme could depict the two inputs: PHD2 is decreased or hypoxia in parallel and the downstream cascade: increased HIF1 α , decreased CAF activation, and eventually decreased # of metastasis.
- The authors should ensure proper spelling of key molecules (in particular in EV figures) and have the manuscript proofread to correct grammatical errors. For example, in Figure EV2, periostin is misspelled periostin, Figure EV4, it is Picrosirius red (and not Picrosirir) and Endomucin (and not Enodmucin), "...CAF-induced ..." should be hyphenated (see Figures EV1, EV2, EV3).

Referee #2:

The revised version properly addresses previous issues. The new experiments of Fig. 7 showing an

impact of CAF PHD2 down-regulation on metastasis is an important addition to the study.

Referee #3:

The authors have addressed the majority of my concerns. However, some minor issues are still present in the manuscript.

The authors refer to Figure 2D in page 6, but it should read Figure 2C.

No error bars in Figure 7A, EV1D and EV2C.

An intriguing issue is the data supplied in the response to reviewers where PHD isoform mRNA is analysed in hypoxia. PHD2 and PHD3 are both hypoxia inducible while PHD1 is not, however, in the data presented by the authors PHD1 is being induced while PHD3 remains flat. Is this correct or was there a mislabelling?

2nd Revision - authors' response

04 August 2015

Answers to referee report_ EMBOR-2015-40107V2:

General response:

We were pleased that the reviewers found our revision satisfactory. Once again, we would like to thank the reviewers for their diligent reading of our work. A detailed point-by-point response to the reviewers' comments is provided below.

Referee #1:

The revised version of the manuscript by Madsen et al. is very satisfactory. The experiment showing that the co-injection of 4T1 tumor cells with shPDH2 CAFs reduced the additive effect on metastasis formation observed when co-injecting tumor cells with CAFs as compared to tumor cells alone (Figure 7B) is a significant improvement. Pending very minor corrections (see list below), I believe that the revised version the manuscript is suitable for publication in EMBO Reports.

Minor corrections:

- The list of authors has been changed since the initial submission. The authors should update accordingly the "Author contribution" section of the manuscript.

We have now corrected the Author contribution.

- The legend of Figure 3 should be relabeled: what is currently labeled G in the text likely corresponds to Figure 3F.

We have now corrected the figure legend for figure 3.

- Figure 7D is still confusing (misleading?), especially the inhibiting mark from PHD2 to HIF-1 α . Instead of the attempt at using a color code, could the authors indicate with up or down arrows the direction of the regulation? Maybe the scheme could depict the two inputs: PHD2 is decreased or hypoxia in parallel and the downstream cascade: increased HIF1 α , decreased CAF activation, and eventually decreased # of metastasis.

We have tried to make the model more intuitive. We have taken away the colours and introduced green arrows to indicate induction of activity and red inhibitory marks to indicate inhibition of activity. The reviewer questions the inhibitory mark from PHD2 to HIF1a, however it is very well established that PHD2 activity targets HIF1a for degradation (see references; Epstein et al, C. elegans EGL-9 and mammalian homologs define a family of dioxygenases that regulate HIF by prolyl hydroxylation. 2001, Cell 107: 43-54 and Wong et al. Emerging novel functions of the oxygen-sensing prolyl hydroxylase domain enzymes. 2013, Trends Biochem Sci 38: 3-11. Just to clarify; in normoxic conditions, oxygen maintains the enzymatic activity of the PHD molecules leading to HIF1a degradation. While in hypoxia the lack of oxygen prevents the activity of PHD proteins thus reducing the hydroxylation of HIF1a and preventing its subsequent degradation. The confusion may be related to the fact that hypoxia can induce PHD2 expression in order to compensate for its lack of enzymatic activity under low oxygen. However, our data clearly demonstrate that hypoxia induces HIF1a stability for up to 72 hours (thus the lack of PHD2 activity dominates over any small induction in its levels). PHD2 depletion in normoxic conditions also promotes HIF1a levels, underlining the fact that the activity of PHD2 is not present under low oxygen.

- The authors should ensure proper spelling of key molecules (in particular in EV figures) and have the manuscript proofread to correct grammatical errors. For example, in Figure EV2, periostin is misspelled periostin, Figure EV4, it is Picrosirius red (and not Picrosirius) and Endomucin (and not Enodmucin), "...CAF-induced ..." should be hyphenated (see Figures EV1, EV2, EV3).

We apologies for the misspelling and we have now corrected all the figures including hyphenation.

Referee #2:

The revised version properly addresses previous issues. The new experiments of Fig. 7 showing an impact of CAF PHD2 down-regulation on metastasis is an important addition to the study.

We thank the reviewer for his support.

Referee #3:

The authors have addressed the majority of my concerns. However, some minor issues are still present in the manuscript.

The authors refer to Figure 2D in page 6, but it should read Figure 2C.

We have now corrected this.

No error bars in Figure 7A, EV1D and EV2C.

We have now added error bars to EV1D and EV2C. We have not added error bars to Fig 7A as this is knockdown efficiency of pool of CAFs. The experiment is a technical replicate, not a biological replicates, therefore we feel that error bars are not appropriate.

An intriguing issue is the data supplied in the response to reviewers where PHD isoform mRNA is analysed in hypoxia. PHD2 and PHD3 are both hypoxia inducible while PHD1 is not, however, in the data presented by the authors PHD1 is being induced while PHD3 remains flat. Is this correct or was there a mislabelling?

The data presented to the reviewers in the rebuttal are correct.

To summarize: hypoxia induces PHD1&2 mRNA expression in CAFs, while PHD3 mRNA levels are unchanged. These data are different from the study by Appelhoff et al, 2004 JBC, 'Differential Function of the Prolyl Hydroxylases PHD1, PHD2, and PHD3 in the Regulation of Hypoxia-inducible Factor', where they show that PHD2&3 protein levels are induced by hypoxia but not PHD1 depending on the cancer cell line they use. The likely reason for the discrepancy is that we measure mRNA levels and not protein levels. It is also important to

state that the significant induction of PHD1 mRNA that we observe comes relatively late (48-72h).

3rd Editorial Decision

04 August 2015

I am very pleased to accept your manuscript for publication in the next available issue of EMBO reports.

Thank you for your contribution to EMBO reports and congratulations on a successful publication.